

# Characteristics of mountain glaciers in the northern Japanese Alps

Kenshiro Arie[1], Chiyuki Narama[2], Ryohei Yamamoto[1,3], Kotaro Fukui[4], Hajime Iida[4]

[1]Graduate School of Science and Technology, Niigata University,8050 Igarashi2-Cho, Nishi-Ku, Niigata, 950-2181
[2]Program of Field Research in the Environmental Sciences, Niigata University,8050 Igarashi2-Cho, Nishi-Ku, Niigata, 950-2181
[3]Aero Asahi Corporation Spatial Information Infrastructure, 50-1165
[4]Tateyama Caldera Sabo Museum, 68 Ashikuraji-bunazaka, Tateyama-machi, Toyama, 930-1405

Correspondence to: Kenshiro Arie (kenshiroarie@gmail.com)

**Abstract.** In 2012, three perennial snow patches in the northern Japanese Alps were determined to be very small glaciers (VSGs: <0.5km$^2$). These were soon followed by four more nearby. However, it had not been determined how such glaciers could be maintained in such a warm climate. In this study, we calculate annual and seasonal mass balances of five of these VSGs, covering 2015–2019 for four of them (2017–2019 for the fifth) using multi-period digital surface models (DSMs) based on structure from motion–multi-view stereo (SfM–MVS) technology and images taken from a small airplane.

The results indicate that, due to mass acquired from avalanches, these VSGs are maintained by acquiring a winter balance that is more than double that from the snowfall amount, thereby exceeding the summer balance. Therefore, we classify them as topographically controlled VSGs. We find almost no annual fluctuation in their summer balance; however, their winter balance, and annual balance, have large annual fluctuations. The annual balance, which mainly depends on the winter balance, showed accumulation throughout each glacier during heavy snow years and ablation throughout each glacier during light snow years. This characteristic differs from the upper accumulation area and lower ablation area that exists on most glaciers. These VSGs had negative annual balance gradients, which suggests that they did not have an equilibrium line during the observation period. Moreover, comparing to other glaciers worldwide, we find the mass balance amplitude of glaciers in the northern Japanese Alps to be the highest measured to date.

## 1 Introduction

More than 100 perennial snow patches are distributed throughout the northern Japanese Alps (Higuchi et al., 1980), with some containing glacial ice as determined by the density (at least 830 kg m$^{-3}$) (Sakita, 1931; Ogasahara, 1964; Tsuchiya, 1978; Kawashima et al., 1993; Kawashima, 1997). Since the 1960s, researchers have tried to determine which ones are glaciers, but the difficulty of measuring flow stymied the earlier efforts (Fukui et al., 2018). However, recently smaller, more accurate surveying instruments have been developed that allow the measurement of ice thickness using ground-penetrating radar (GPR). Using these instruments together with the cm-scale accuracy of global navigation satellite system (GNSS) surveys, several




groups measured the ice thickness and horizontal flow velocity of seven perennial snow patches in the region, finding them to be active glaciers, subsequently named Gozenzawa, Sannomado, Komado, Kakunezato, Karamatsuzawa, Kuranosuke, and

Ikenotan ( Fukui and Iida, 2012; Fukui et al., 2018; Arie et al., 2019). As they are less than 0.5 km$^2$ in area, they are classified as very small glaciers (VSGs) (Huss, 2010; Huss and Fischer, 2016).

In general, glaciers form and change in response to their mass balance, which is determined by the accumulation of primary snowfall and ablation of primary snow. Therefore, to understand the factors contributing to glacier formation and persistence, one must measure accumulation and ablation (Ohmura, 2010). Accumulation and ablation can be substituted by

winter and summer balances, also called the seasonal balance (Ohmura, 2011). This approach should also apply to VSGs.

The seasonal balance incorporates the geographical characteristics of glaciers, and is crucial for assessing the relationship between the climatic environment and glacier mass balance (Dyurgerov and Meier, 1999; Huss et al., 2008; Ohmura, 2011; Pelto et al., 2019). A related quantity is the mass balance amplitude, which Meier (1984; 1993) defines as half of the sum of the absolute values of the winter and summer mass balances. In the Glossary of Mass Balance and Related Terms

(Cogley et al., 2011), the mass balance amplitude tends to be higher for glaciers in maritime climates than those in continental climates due to the former having higher accumulation.

In the northern Japanese Alps, Fukui et al. (2018) measured the mass balance of the Gozenzawa glacier using the stake, or glaciological, method. Their study indicated that the mass balance in 2012−2015 had accumulation throughout, whereas in 2015−2016 it had ablation throughout. In addition, they showed that avalanches contributed significantly to the

accumulation of Gozenzawa Glacier. However, the stakes were measured only twice, in the autumns of 2012 and 2016, the seasonal balance not being determined. For measuring winter and summer balances, the stake method is not reliable in the northern Japanese Alps because the stakes are buried under heavy snowfall during winter. In some years, they remain buried even at the end of the snowmelt season.

Although some characteristics of VSGs that persist in warm environments at middle latitudes (here: 36.57°–36.69°N)

and low altitudes (here: 1,750–2,770 m) have been explored, the mechanisms by which the VSGs in the northern Japanese Alps are formed and maintained remain unclear. To clarify the mass balance characteristics of these VSGs, we measured the annual and seasonal mass balance of five VSGs in the northern Japanese Alps using a geodetic method. The geodetic method obtains the change in mass by evaluating the elevation change of the entire glacier surface between two dates (Ohmura, 2010, 2011). This method has been used in recent years to measure the mass balance of many glaciers because it can obtain data for

parts of the glacier that are difficult to access and observe on-site.

Here, we calculate the annual and seasonal mass balance of four VSGs in 2015–2019 and for one VSG in 2017–2019 by comparing multi-period digital surface models (DSMs) created using structure from motion–multi-view stereo (SfM–MVS) software and aerial images taken from a small airplane. We also compare the mass balance amplitude and mass balance profile of VSGs in the northern Japanese Alps with those of other glaciers worldwide.



## 2 Study area


The climate of the northern Japanese Alps is greatly influenced by the winter monsoon and Tsushima warm current in the Sea of Japan. During the winter monsoon, a dry cold air mass from the continent passes over the Tsushima warm current, gaining heat and water vapor, then undergoes topographic updraft, producing some of the heaviest snowfall worldwide over the northern Japanese Alps (Nosaka et al., 2019; Kawase et al., 2020). For example, the average snow depth at Tateyama

Murododaira (~2,450 m; Fig. 1) during 1996–2018 was about 8 m (Iida et al., 2018).

In this study, we focus on five VSGs in the northern Japanese Alps (Fig. 1). Briefly, their characteristics and environments are as follows. The Gozenzawa Glacier (Fig. 2a) lies on the eastern side of Mt. Oyama (3,003 m), at the bottom of the north-eastern curve in the Gozenzawa valley. The Sannomado Glacier (Fig. 2b) sits at the bottom of the glacial erosion valley between the Sannomado and Hachimine ridges, and trends east–southeast. The Komado Glacier (Fig. 2b) lies at the

bottom of the glacial erosion valley between the southeast side of Mt. Ikenotaira (2,561 m) and the Sannomado ridge, and trends east–southeast. The Kakunezato Glacier (Fig. 2c) is at the head of the glacial erosion valley extending northeast from the northern peak of Mt. Kashima-Yarigatake (2,842 m). The Karamatsuzawa Glacier (Fig. 2d) lies at the head of the glacial erosion valley extending northeast from the northern peak of Mt. Karamatsu (2,696 m).

The bedrock surrounding the Sannomado and Komado Glaciers is diorite, whereas that for the Gozenzawa and

Kakunezato Glaciers is granite and that for the Karamatsuzawa Glacier is serpentine. These geological differences influence the differing amounts of sediment on the glacier surfaces. In particular, the Sannomado and Komado Glaciers are clean-type glaciers with few debris, whereas the Gozenzawa, Kakunezato, and Karamatsuzawa Glaciers have a debris-covered area in their middle and terminal regions during ablation years.

For further details about these glaciers, see Table 1 (also, Fukui and Iida 2012; Fukui et al. 2018; Arie et al. 2019).



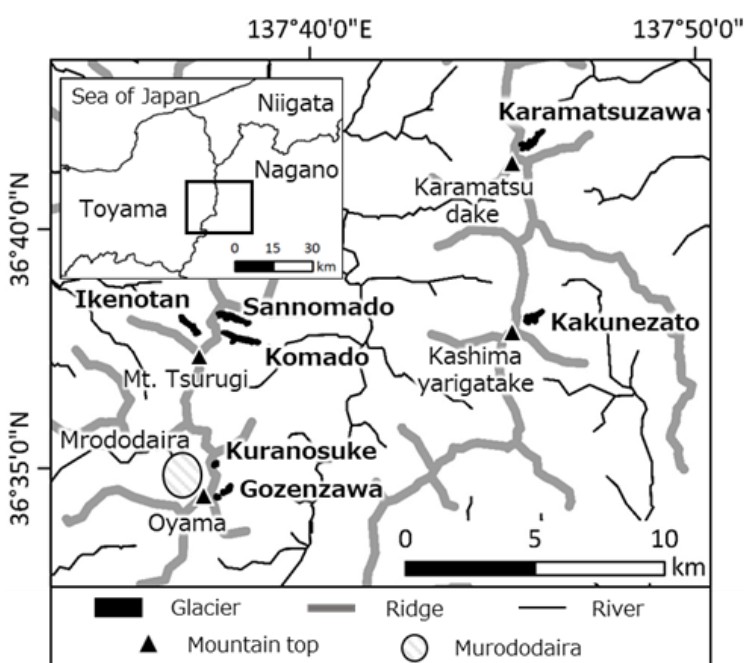

**Figure 1 Seven glaciers in the northern Japanese Alps (names in bold font).**

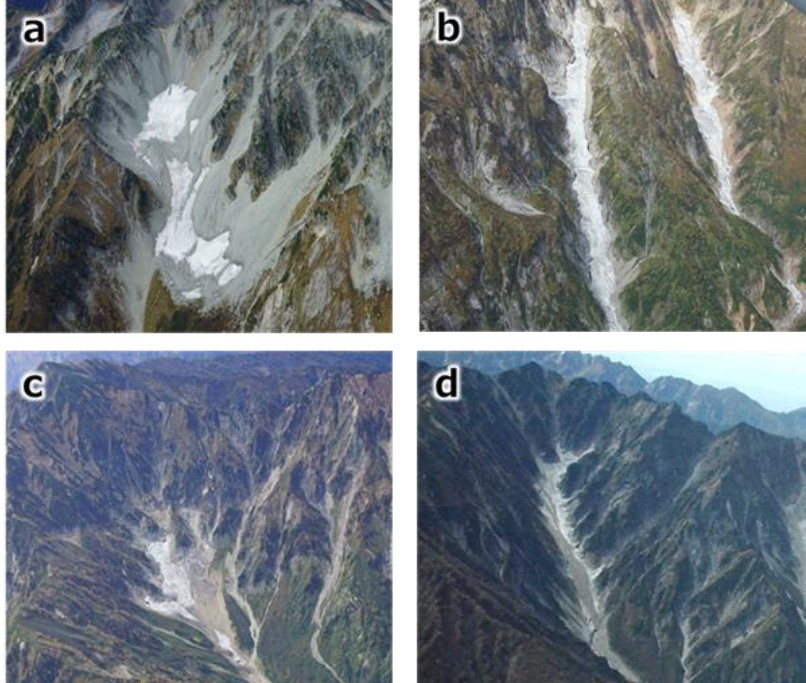

**Figure 2 The five very small glaciers (VSGs) in this study at the end of snowmelt season in 2016. a) Gozenzawa. b) Sannomado (left), Komado (right). c) Kakunezato (left). d) Karamatsuzawa. All taken on Sep. 30.**



**Table 1 Physical properties of the glaciers. Maximum depth and maximum horizontal velocity at Gozenzawa, Sannomado, Komado, Kakunezato Glacier were showed in Fukui et al. (2018).**

| Glacier | Gozenzawa | Sannomado | Komado | Kakunezato | Karamatsuzawa |
|---|---|---|---|---|---|
| Length (m) | 760 | 1420 | 1270 | 740 | 1080 |
| maximum width (m) | 200 | 110 | 210 | 250 | 150 |
| Area (km²) | 0.074 | 0.101 | 0.109 | 0.087 | 1.034 |
| Altitude range (m) | 2510-2770 | 1760-2500 | 1910-2300 | 1800-2150 | 1750-2350 |
| Average inclination (°) | 19.9 | 27.7 | 20.4 | 25.6 | 26.4 |
| maximum depth (m) | 27 | 48 | 30> | 30> | 35 |
| maximum horizontal velocity (m a⁻¹) | 0.63 | 3.65 | 3.77 | 2.39 | 3.15 |

## 3.1 Methods

### 3.1 Data acquisition

We determine the seasonal and annual mass balances of the five glaciers by comparing multi-period DSMs created using aerial images and SfM-MVS software. The images were taken under clear weather with few clouds from a small Cessna aircraft at the end of the snowmelt season (late September to early October) and during the maximum snow depth season (late March to early April) from 2015 to 2019. The cameras were a Sony α7II (24.3 million pixels) from 2015 (end of snowmelt season) to 2018 (maximum snow depth season) and a Sony α7RII (42.4 million pixels) from 2018 (end of snowmelt season) to 2019 (end

of snowmelt season). Images were taken every 2 s to obtain a complete view of the entire glacier and surrounding terrain at the altitude range of 3,500–3,800 m. The AUTO mode was used from 2015 (end of snowmelt season) to 2017 (maximum snow depth season). From 2017 (end of snowmelt season) to 2019 (end of snowmelt season), we used a shutter setting less than 1/1600 s, and an ISO of 100–200, with the F value set to automatic. The flight route first visits Gozenzawa, then Sannomado, Komado, Karamatsuzawa (from 2017), and Kakunezato Glaciers.

The observation period included both light and heavy snow years. For 1996–2018, the measured snow depths at Tateyama Murododaira (2,450 m) were lowest in 2016 and second-highest in 2017 (Iida et al., 2018).

### 3.2 SfM-MVS analysis

SfM is a calculation technique that allows automatic camera position determination in three-dimensional (3D) space. After

estimating the camera positions, we use additional dense image-matching algorithms such as MVS to calculate the dense 3D point cloud of the surveyed object on an arbitrary relative scale (Piermattei et al., 2015). This relative scale must then be converted into an absolute scale to obtain geometric measurements using real-world coordinates or a known field distance (Dai et al., 2014).

Creating a DSM is as follows. First, high-density point cloud data are created from continuous aerial images using

SfM-MVS. Next, multiple (three or more) ground control points (GCPs) are defined in the created point cloud data and



geometrically corrected. Finally, these corrected data are used to create a DSM. Setting the GCPs is the only manual work in this process because Pix4dmapper automates the first and last steps.

We set the GCPs using a DSM (resolution: 0.5 m) created from both aerial laser survey data (from the Ministry of Land, Infrastructure, Transport and Tourism) and orthophoto-corrected images of aerial images obtained at the time of the

survey. The error in the height direction of the aerial laser survey data is 0.4–0.6 m in a region with large undulations (Sato et al., 2004). The aerial laser survey data are from the end of the snowmelt season in 2009 for the Sannomado and Komado Glaciers, in 2010 for the Kakunezato Glacier, in 2011 for the Gozenzawa Glacier, and in 2014 for the Karamatsuzawa Glacier. The GCPs were positioned to surround the glacier. At the end of the snowmelt season, the GCPs were set at long-term immovable and easy-to-read buildings and rocks. In the maximum snow depth season, GCPs were set at an exposed rock wall

without snow at the same location for each image. Figure 3 shows the locations of GCPs for each glacier and period. The numbers of GCPs for each glacier at the end of the snowmelt season are 17 at Gozenzawa Glacier, 28 at Sannomado and Komado Glaciers, 17 at Kakunezato Glacier, and 17 at Karamatsuzawa Glacier. For the maximum snow depth season, these numbers are instead 5, 6, 5, and 7, respectively.

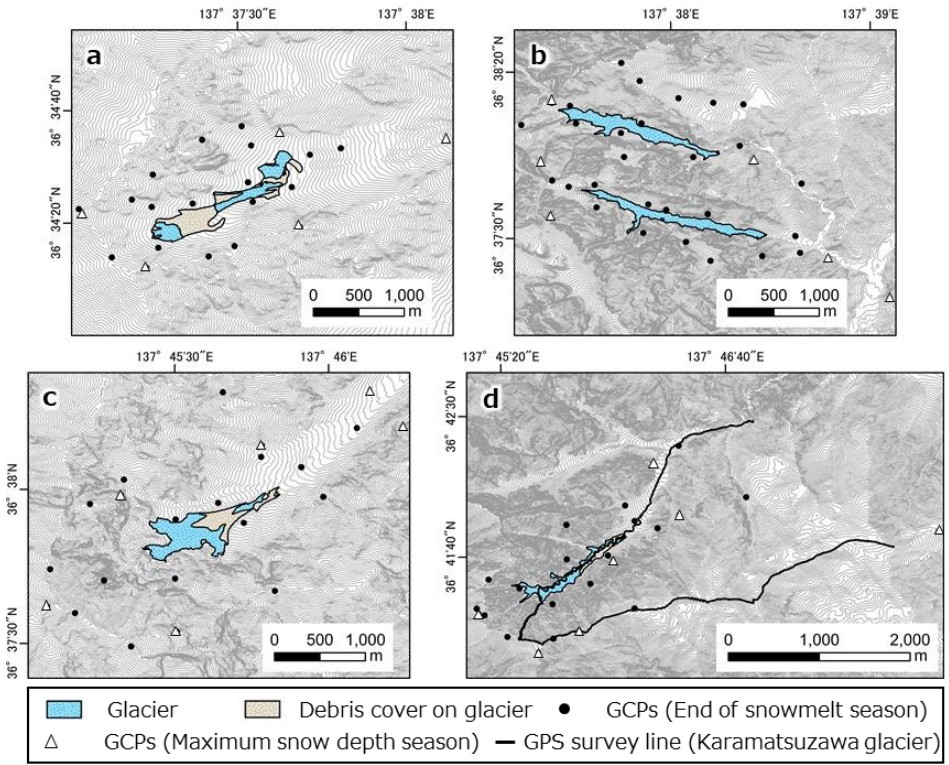

**Figure 3 Glacier area and GCP locations for each glacier and season. a) Gozenzawa. b) Sannomado (lower), Komado (upper). c) Kakunezato. d) Karamatsuzawa. Black line is the GPS survey path. Contour interval is 10 m.**




### 3.3 Geodetic mass balance determination

In the geodetic method, the mass balance of glaciers is estimated by the change in volume as judged by comparing the DSMs for two periods and multiplying the result by the estimated ice density (Huss, 2013; Piermattei et al., 2016). This estimated relative volume change equals the surface mass balance plus the vertical component of glacier flow (i.e., the emergence velocity). Therefore, to express the surface mass balance, one integrates the relative volume change for the entire glacier to remove the vertical flow component (Ohmura, 2010). The resulting mass balance B is

$$B = (\delta V \times \rho)/A \; , \qquad\qquad (1)$$

where $B$ is expressed in m of water equivalent (m weq), $\delta V$ is the relative volume change (m³), $\rho$ is the ice density (kg m⁻³), and $A$ is the glacier area (m²). To determine $\delta V$, we compare DSMs from the end of one snowmelt season to the end of the next. The area A is determined at the end of snowmelt season, an approach called a stratigraphic system (Unesco/IASH., 1970). Here, differences between DSMs were calculated using a geographic information system (GIS). For the ice density, the average value for Gozenawa was 695 kg m⁻³ in the top 0.7 m (570–740 kg m⁻³) and 860 kg m⁻³ at depths below 0.7 m (824–907 kg m⁻³) at the end of the snowmelt season (Fukui et al., 2018). In the perennial snow patches of Japan, snow transitions to glacier ice during a one-year period (Kawashima 1997); therefore, the layer between the snow surface and a depth of 0.7 m consists of residual snow from the previous winter, whereas the deeper layer consists of ice formed earlier. For annual balance calculations for all glaciers, we used a snow and ice density of 695 kg m⁻³ if the balance was positive and 860 kg m⁻³ if the balance was negative.

We also compared DSMs created for the maximum snow depth season with those for the end of the snowmelt season, and calculated the change in relative volume during the accumulation and ablation seasons. We calculate the winter and summer balances of the glaciers by multiplying the snow density for the maximum snow depth season with the calculated volume change. At Tateyama Murododaira, Iida et al. (2018) measured the density in the snow-cover cross-sections in late March 1996–2018, getting an average of 431 kg m⁻³, a value we use to calculate the seasonal mass balance.

We apply these densities to the annual and seasonal mass balances for the Gozenzawa, Sannomado, Komado, and Kakunezato Glaciers in 2015−2016, 2016−2017, 2017−2018, and 2018−2019, as well as for the Karamatsuzawa Glacier in 2017−2018 and 2018−2019. For each glacier, the mass balance calculation uses the glacier area when this area was smallest. This occurred on October 16, 2019, for Gozenzawa, Sannomado, Komado, and Karamatsuzawa Glaciers and on September 30, 2016, for Kakunezato Glacier.





### 3.4 Data accuracy

Following the method in Immerzeel et al. (2014), we estimate the error for the annual balance at the end of the snowmelt season by first creating a 10-m-wide buffer zone around the outside of the glacier area on the base area (Fig. 4). Then, we calculate the geodetic annual balance error as the mean and standard deviation (SD) of the altitude difference between the buffer zone.

However, because the base area is covered with snow during winter, we must use a different method to estimate the annual balance error. For this estimate, we ran kinematic GPS surveys of Karamatsuzawa Glacier and the ridges of Mts. Karamatsu and Happoike-Sanso one day after the aerial photography (Fig. 3). We then equate the geodetic seasonal mass balance error as the SD of the altitude difference between the measured kinematic GPS data for March 19 and the DSM of Karamatsuzawa Glacier on March 18, both in 2019. The GPS surveying instrument was GEM-1 (Enabler; formerly GNSS Technologies), with calculated coordinates of the GPS antenna and post-processing using the open-source program RTKLIB (ver. 2.4.3).

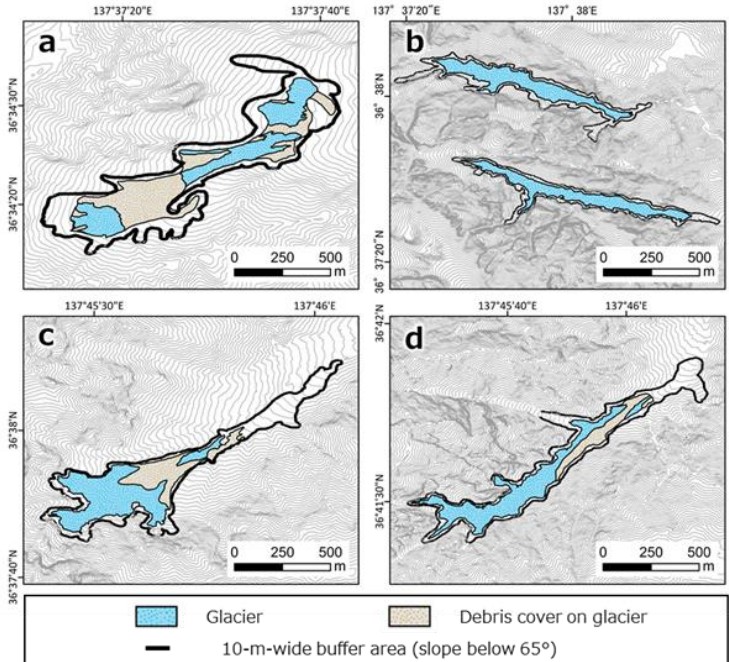

**Figure 4 Glacier area and 10-m-wide buffer zone used for uncertainty estimate. a) Gozenzawa. b) Sannomado and Komado. c) Kakunezato. d) Karamatsuzawa. Contour interval is 10 m.**

### 3.5 Mass balance amplitude

The mass balance amplitude ($\alpha$) equals the average of the absolute values of the winter ($Bw$) and summer ($Bs$) balances:





$$\alpha = (|Bw| + |Bs|)/2 \ . \tag{2}$$

We use this equation to determine the mass balance amplitude for the Gozenzawa, Sannomado, Komado, and Kakunezato Glaciers, where the winter and summer balances were measured during 2015–2019. We then compare these four-year averages to the average mass balance amplitude of other glaciers worldwide. The worldwide glaciers were all glaciers

observed for periods longer than 5 years according to the "fluctuations of glaciers" database of the World Glacier Monitoring Service (World Glacier Monitoring Service (WGMS), 2020). Glacier locations are shown in Fig. 5.

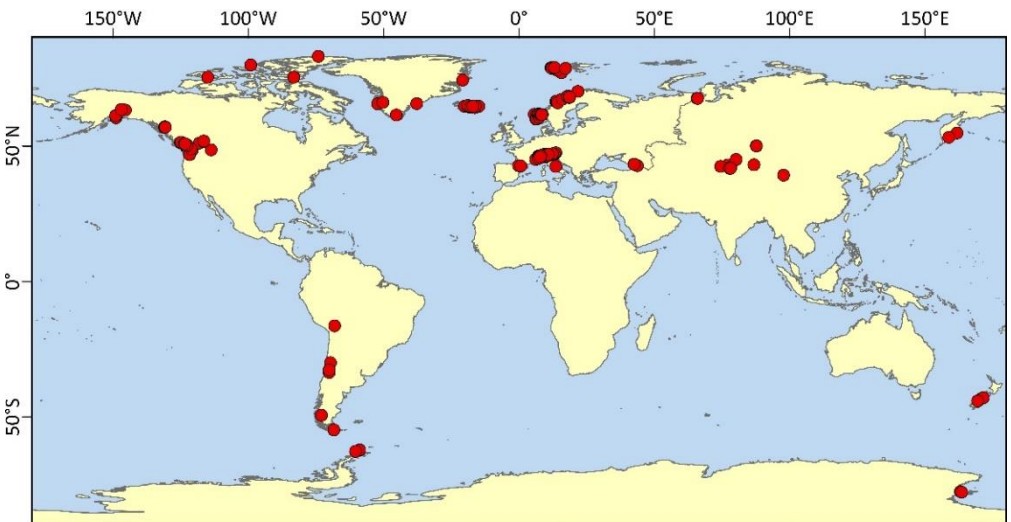

**Figure 5 Glacier locations used for comparison of winter and summer balances (from WGMS, 2020). All glaciers had been observed for periods longer than 5 years except those in Japan. Each circle represents the following numbers. Alps, 51 glaciers; Andes, 7 glaciers; Antarctic, 3 glaciers; Arctic North America, 3 glaciers; Caucasus, 3 glaciers; Dry Valley, 2 glaciers; Greenland, 5 glaciers; high mountain Asia (HMA: Tien Shan, Altai, Qilian), 13 glaciers; Iceland, 12 glaciers; Kamchatka, 2 glaciers; New Zealand, 3 glaciers; Scandinavia, 38 glaciers; Svalbard, 9 glaciers; Urals, 2 glaciers; western North America, 27 glaciers.**

**3.6 Mass balance gradient and emergence velocity**

On many glaciers, the amount of ablation and accumulation vary systematically with altitude and thus the mass balance has

an altitude profile (Benn and Evans, 2014). In general, the amount of ablation and accumulation is increasing with altitude due to the decrease in temperature.

We divided each glacier region into 10-m altitude intervals and calculated the mass balance profile (annual, winter, summer) using the mass balance of each interval. This was done for the Gozenzawa, Sannomado, Komado and Kakunezato Glaciers because they were observed for four years. In addition, altitudinal differences calculated by the geodetic method

include both the surface mass balance and the emergence velocity (the vertical flow component). Thus, calculating the mass balance profile involves removing the influence of emergence velocity. After obtaining the mass balance profile, we derived the mass balance gradient based on mass balance profile.



The emergence velocity ($Ve$) is expressed using the flux method as

$$Ve = (Q\text{in} - Q\text{out})/(W \times x), \qquad (3)$$

where $Q$ is the ice flux into and out of the target area, $W$ is the average glacier width, and $x$ is the longitudinal length of the target area (Nuimura et al., 2011).

The ice flux at the boundaries of the target area is


$$Q = W \times h \times v, \qquad (4)$$

where $W$, $h$, and $v$ are the glacier width, depth, and flow velocity (Nuimura et al., 2011). For the Karamatsuzawa Glacier, we measured $W$ and $v$, and use the depth measurement from an earlier study (Arie et al., 2019). For the flow $v$, we measured the
stake locations on 22 October, 2019 using a GEM-3 GNSS surveying instrument (Enabler) and then calculated the annual flow by comparing to their locations in 2018 (Arie et al., 2019). The earlier study had inserted 4.6-m-long stakes at five points on the glacier (Fig. 6, P1–5) and used GNSS surveyors to measure the stake locations on 23 October, 2018 (As a check, we also compared a base point location near the glacier terminus (Fig. 6, P6) between 2018 and 2019). With the five flow measurements, we determine $Ve$. We then use the maximum value as the mass balance profile error for four glaciers.


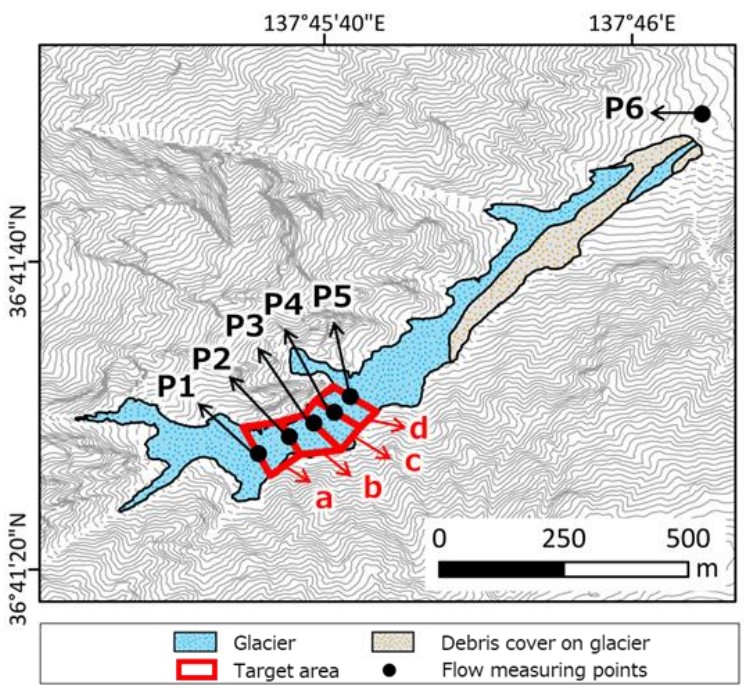

**Figure 6 Target area for emergence velocity estimation and flow measurement point in the Karamatsuzawa glacier. Contour interval is 10 m.**





# 4 Results

## 4.1 Data accuracy

For the different DSMs, the SD of differences in slope within the 10 m buffer zone around the glacier area increases with the slope of the terrain, particularly above 60°, as shown in Fig. 7a. Such an increase is consistent with the finding of Piermattei

et al. (2016). However, regions with a slope below 65° account for over 75% of the total buffer area (Fig. 7b).

Thus, we evaluated the value in the region of the polygon with an inclination of 65° or less. Table 2 shows that all mean and SD values are below 1 m, except for 2018-2019 on Komado Glacier where the SD is 1.11 m. As the glacier slopes and buffer regions sloped less than 65°, the annual balance calculation method has sufficient accuracy.

To evaluate DSM accuracy during the maximum snowfall season, the day after obtaining aerial images for this season,

we obtained surface altitude data via a kinematic GPS survey. The survey was done when a researcher attached a GPS antenna to their pack, then walked the ridge between Karamatsudake and Happoike Sanso, then snowboarded down Karamatsuzawa Glacier. The mean and SD of the altitude differences between the Karamatsuzawa Glacier DSM (March 18, 2019) and that from the GPS data (March 19, 2019) are 1.89 and 1.73 m, respectively. Assuming an average height of 1.5 m from the ground to the antenna (on the pack), the mean and SD of the altitude difference between the DSM and GPS survey data are 0.39 and

1.73 m, respectively. We used this SD value as the altitude-difference error for the winter and summer seasons. Then, using 431 kg m$^{-3}$ as the mean ice density (late March value at Tateyama Murododaira), the calculated error between the winter and summer balances is ± 0.75 m weq.

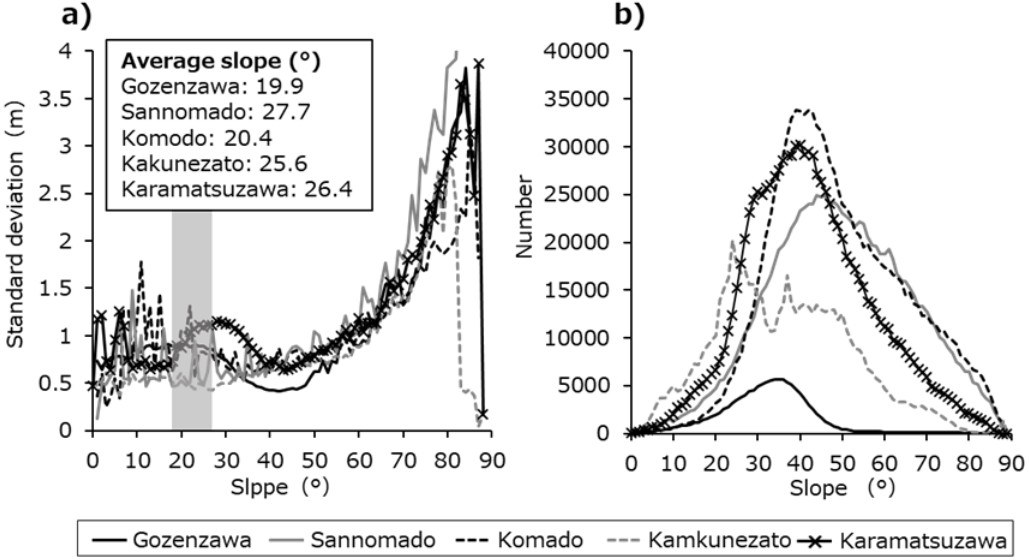

**Figure 7 Comparison of the DSMs for the end of the 2017 and 2018 end of snowmelt seasons. a) The standard deviation SD of differences in slope between the different DSMs used. b) The number of pixels within the 10 m buffer around the glacier area. Shaded band along abscissa axis is the range of average slopes of the five glaciers.**



**Table 2 Mean and SD of the altitude differences in the DSMs for the glacier buffer regions with an inclination of 65° or less in the given years. In parentheses are the mean slopes in the 10-meter-wide polygon around each glacier.**

| Year | Gozenzawa (31°) | | Sannomado (44°) | | Komado (43°) | | Kakunezato (36°) | | Karamatsuzawa (42°) | |
|---|---|---|---|---|---|---|---|---|---|---|
| | Mean | SD | Mean | SD | Mean | SD | Mean | SD | Mean | SD |
| 2015-2016 | 0.1 | 0.55 | -0.81 | 0.96 | -0.66 | 0.87 | -0.6 | 0.85 | | |
| 2016-2017 | -0.15 | 0.3 | 0.38 | 0.99 | 0.37 | 0.73 | -0.12 | 0.79 | | |
| 2017-2018 | -0.04 | 0.33 | -0.06 | 0.9 | -0.01 | 0.81 | -0.33 | 0.62 | -0.07 | 0.92 |
| 2018-2019 | 0.2 | 0.29 | -0.37 | 0.85 | -0.08 | 1.11 | -0.28 | 0.74 | 0.24 | 0.88 |
| Average | 0.03 | 0.37 | -0.22 | 0.93 | -0.10 | 0.88 | -0.33 | 0.75 | -0.07 | 0.92 |

## 4.2 Mass balance

For all glaciers, the annual balance is negative in all years except 2016-2017 (Table 3), a heavy snow year. All glaciers are also consistent in all having their largest mass reduction in 2015-2016, a light snow year. In this year, the ablation area extends throughout each glacier (first column, Fig. 8), whereas in the heavy snow year that followed (2016-2017), each glacier is entirely an accumulation area (second column, Fig. 8). These characteristics agree with Fukui et al.'s (2018) findings for the Gozenzawa Glacier.

For the winter and summer balances, the values for all glaciers are 5-11 m weq, as shown in Table 4. We can compare this value to the average snow depth of 6.8 m on Tateyama Murododaira (1996–2018) (Iida et al., 2018). Considering that the average value of the mean density here is 431 kg m$^{-3}$, this snow amount is 2.93 m weq. Unlike the glaciers, the terrain at Tateyama Murododaira (2,450 m) is flat and not affected by the topographical influence of avalanches and snowdrifts. But in the northern Japanese Alps, the glaciers had a winter balance of 5-11 m weq, or over 2-4 times as much as the snow amount

at Tateyama Murododaira. Presumably, this additional winter balance is from avalanches and snowdrifts.

Between the light and heavy snow years, the winter balance increases by about 3-5 m weq (Table 4), whereas their summer balance remains about the same. This difference in behaviour between summer and winter can be seen to occur throughout the four-year period in the plots in Fig. 9. Namely, the summer balance remains nearly constant compared to the variations in the annual and winter balances.

The cumulative mass balances are shown in Fig. 10. From 2015 to 2019, the trends show significant mass loss, decreasing by 4-5 m weq for the Gozenzawa and Kakunezato Glaciers, which had debris cover. For the for Sannomado and Komado Glaciers, which did not have debris cover, the loss is larger, at 7-9 m weq. The Karamatsuzawa Glacier also has significant mass loss, though the period covered is only three years (2017-2019).





**Table 3 Annual mass balance of each glacier (surface area in parenthesis) calculated from the DSM at the end of snowmelt season.**

**Gozenzawa (74093m²)**

| Year | Relative altitude change (m) | | Relative volume change (m³) | | Annual mass balance (m weq) | |
|---|---|---|---|---|---|---|
| | Original | Correction | Original | Correction | Original | Correction |
| 2015-2016 | -3.46 | -3.56±0.55 | -256362 | -263771±40751 | -2.98 | -3.06±0.47 |
| 2016-2017 | 1.88 | 2.03±0.30 | 139295 | 150408±22228 | 1.31 | 1.41±0.21 |
| 2017-2018 | -1.80 | -1.76±0.33 | -133367 | -130403±24451 | -1.55 | -1.51±0.28 |
| 2018-2019 | -1.30 | -1.50±0.29 | -96321 | -111139±21487 | -1.12 | -1.29±0.25 |

**Sannomado (101040m²)**

| Year | Relative altitude change (m) | | Relative volume change (m³) | | Annual mass balance (m weq) | |
|---|---|---|---|---|---|---|
| | Original | Correction | Original | Correction | Original | Correction |
| 2015-2016 | -8.01 | -7.2±0.96 | -809330 | -727488±96998 | -6.89 | -6.19±0.83 |
| 2016-2017 | 3.53 | 3.15±0.99 | 356671 | 318276±100030 | 2.45 | 2.19±0.69 |
| 2017-2018 | -1.13 | -1.07±0.90 | -114175 | -108113±90936 | -0.97 | -0.92±0.77 |
| 2018-2019 | -3.72 | -3.35±0.85 | -375869 | -338484±85884 | -3.20 | -2.88±0.73 |

**Komado (109390m²)**

| Year | Relative altitude change (m) | | Relative volume change (m³) | | Annual mass balance (m weq) | |
|---|---|---|---|---|---|---|
| | Original | Correction | Original | Correction | Original | Correction |
| 2015-2016 | -7.85 | -7.19+0.87 | -858704 | -786507±95169 | -6.75 | -6.18±0.75 |
| 2016-2017 | 2.67 | 2.3±0.73 | 292069 | 251595±79854 | 1.86 | 1.60±0.51 |
| 2017-2018 | -0.82 | -0.81±0.81 | -89699 | -88605±88606 | -0.71 | -0.70±0.70 |
| 2018-2019 | -4.02 | -3.94±1.11 | -439744 | -430993±121423 | -3.46 | -3.39±0.95 |

**Kakunezato (86870m²)**

| Year | Relative altitude change (m) | | Relative volume change (m³) | | Annual mass balance (m weq) | |
|---|---|---|---|---|---|---|
| | Original | Correction | Original | Correction | Original | Correction |
| 2015-2016 | -5.71 | -5.11±0.85 | -496028 | -443905±73836 | -4.91 | -4.39±0.73 |
| 2016-2017 | 3.54 | 3.54±0.79 | 307520 | 317944±68627 | 2.46 | 2.54±0.79 |
| 2017-2018 | -2.19 | -2.19±0.62 | -190245 | -161578±53859 | -1.88 | -1.59±0.53 |
| 2018-2019 | -1.92 | -1.64±0.74 | -166790 | -142466.8±64283 | -1.65 | -1.41±0.64 |

**Karamatsuzawa (103400m²)**

| Year | Relative altitude change (m) | | Relative volume change (m³) | | Annual mass balance (m weq) | |
|---|---|---|---|---|---|---|
| | Original | Correction | Original | Correction | Original | Correction |
| 2017-2018 | -1.27 | -1.01±0.92 | -131318 | -104434±95128 | -1.09 | -0.87±0.79 |
| 2018-2019 | -2.08 | -1.82±0.88 | -215072 | -188188±90992 | -1.79 | -1.57±0.76 |

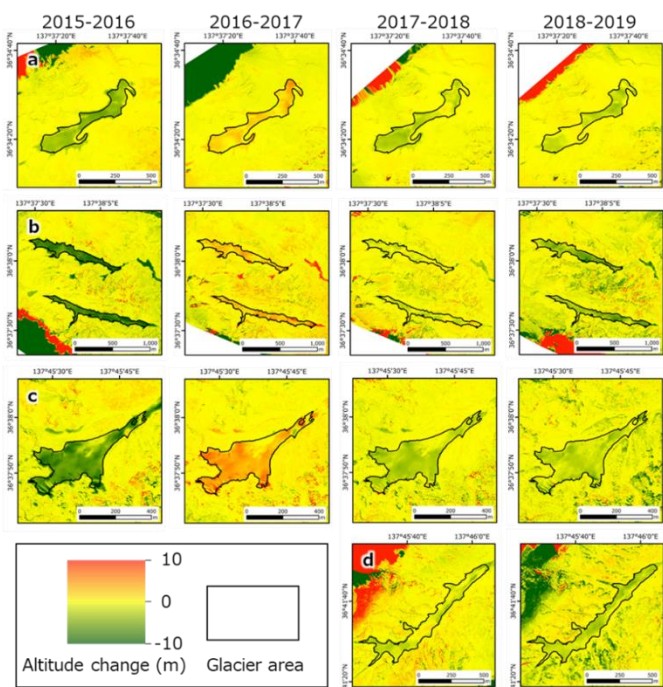

**Figure 8 Annual altitude changes of each glacier and year. a) Gozenzawa. b) Sannomado (lower), Komado (upper). c) Kakunezato. d) Karamatsuzawa.**





**Table 4 Winter and summer balances (m weq).**

| Year | Gozenzawa | | Sannomado | | Komado | | Kakunezato | | Karamatsuzawa | |
|---|---|---|---|---|---|---|---|---|---|---|
| | Winter | Summer | Winter | Summer | Winter | Summer | Winter | Summer | Winter | Summer |
| 2015-2016 | 5.63 | 7.16 | 7.52 | 10.97 | 6.65 | 10.04 | 8.38 | 10.59 | | |
| 2016-2017 | 8.96 | 8.09 | 12.43 | 10.90 | 10.44 | 9.29 | 12.72 | 11.19 | | |
| 2017-2018 | 7.64 | 8.42 | 9.62 | 10.10 | 9.37 | 9.72 | 9.95 | 10.89 | 10.15 | 10.63 |
| 2018-2019 | 7.16 | 7.72 | 10.03 | 11.64 | 9.14 | 10.87 | 10.51 | 11.34 | 10.45 | 11.21 |
| Average | 7.35 | 7.85 | 9.90 | 10.90 | 8.90 | 9.98 | 10.39 | 11.00 | 10.30 | 10.92 |
| SD | 1.19 | 0.47 | 1.74 | 0.54 | 1.39 | 0.58 | 1.55 | 0.29 | 0.15 | 0.29 |


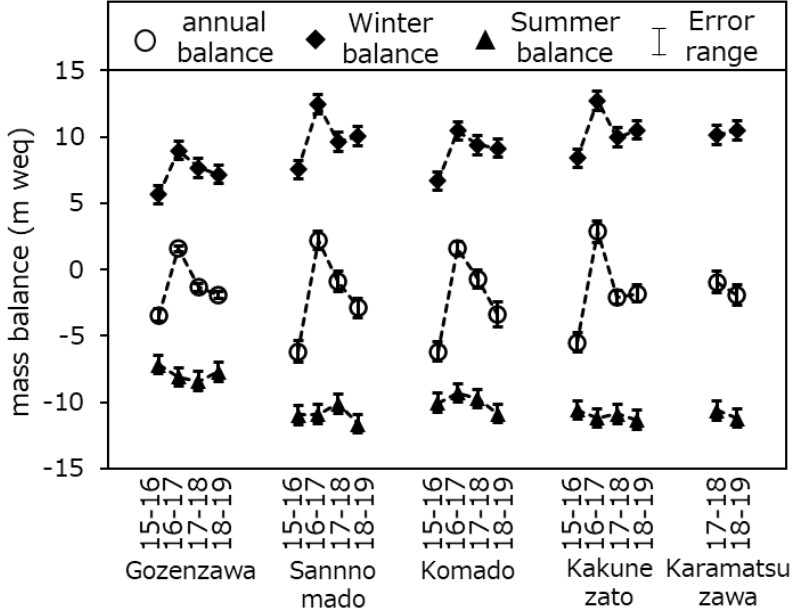

**Figure 9 Annual mass balance, winter balance, and summer balance of each glacier in 2015-2019.**

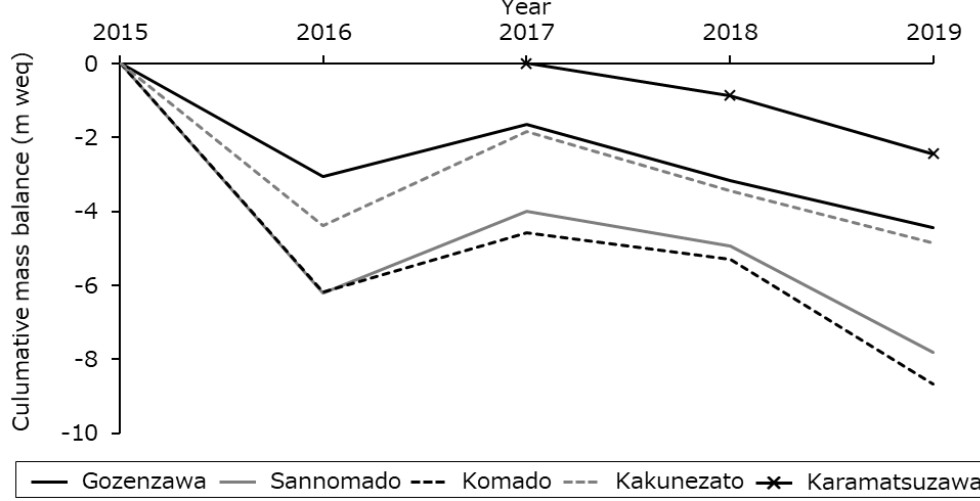

**Figure 10 Cumulative mass balance of each glacier with baseline on 2015 data (2017 for Karamatsuzawa).**



### 4.3 Mass balance amplitude

The mass balance amplitudes of these four glaciers, are much higher than glaciers studied elsewhere. We plot the comparison in Fig. 11. In general, the mass balance amplitudes of glaciers in polar regions and HMA (high mountain Asia) are low, and those for glaciers in maritime climates such as New Zealand and Kamchatka are higher (Cogley et al., 2011). One exception to this trend is the relatively high value for glaciers in the Urals, despite their dry, cold climate. These mass balance amplitudes increase nearly linearly with the SD of the annual balance as shown in Fig. 12.

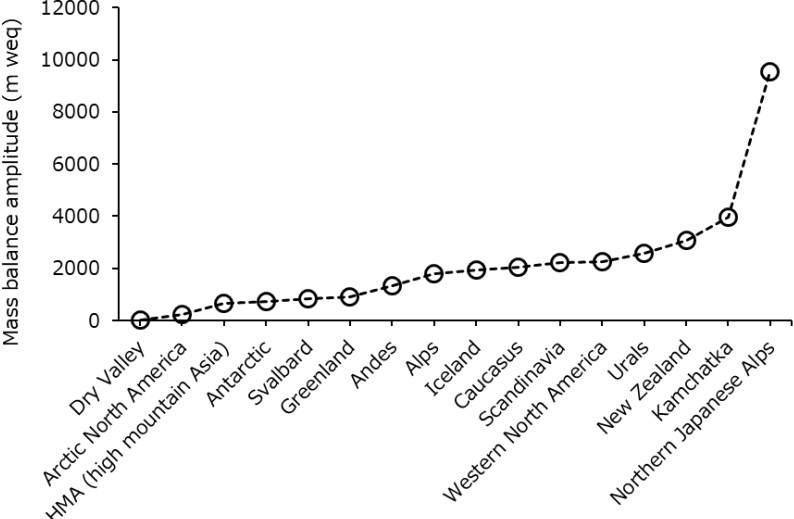

**Figure 11 Mass balance amplitudes of glaciers.**

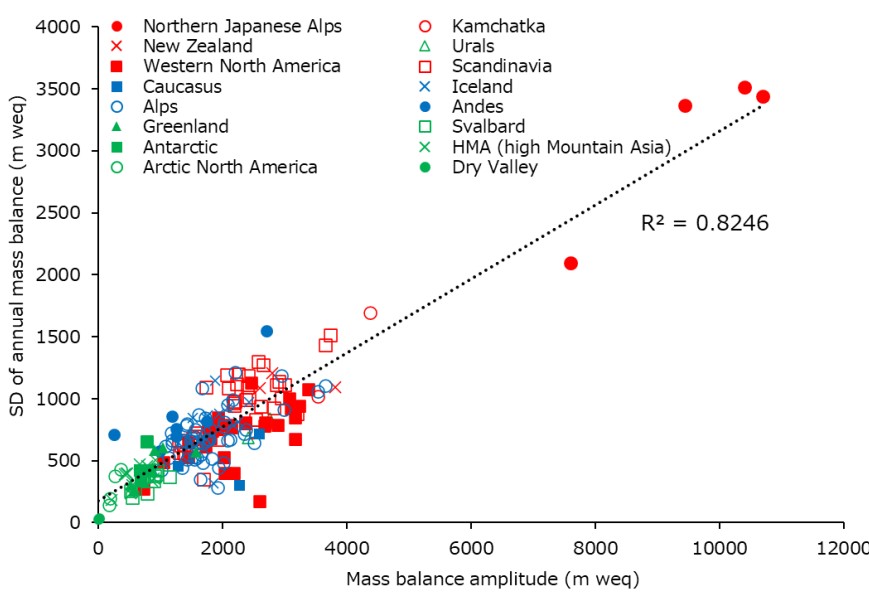

**Figure 12 Standard deviation of annual mass balance versus mass balance amplitude.**



## 4.4 Emergence velocity of Karamatsuzawa Glacier

The measurement error of the GNSS survey was set at 0.07 m, according to annual migration relative to a base point near the glacier terminal. The annual flow of Karamatsuzawa Glacier is shown in Table 5; all measurements exceeded the error level of 0.07, indicating significant flow.

Using the measured results for the Karamatsuzawa Glacier (Table 5), we find the emergence velocity in the different target areas to range from –0.04 to 0.16 m/a. For determining uncertainty in the mass balance profiles, we use the maximum value of 0.16 m/a, giving an uncertainty of 0.16 m.

**Table 5 Analyses results of the Karamatsusawa Glacier. Regions refer to Fig. 6.**

|  | P1 | P2 | P3 | P4 | P5 | a | b | c | d |
|---|---|---|---|---|---|---|---|---|---|
| Flow Velocity (m a-1) | 1.90 | 2.32 | 2.03 | 2.14 | 2.20 | | | | |
| Ice thickness (m) | 28 | 35 | 28 | 27 | 28 | | | | |
| Width (m) | 106 | 72 | 92 | 90 | 86 | | | | |
| longitudinal length of target area (m) | | | | | | 60 | 47 | 40 | 40 |
| Mean width of target area (m) | | | | | | 89 | 82 | 91 | 88 |

## 4.5 Mass balance gradient

We now examine the mass-balance profiles of the four VSGs studied during 2015–2019. Consider the gradient trends of these profiles in summer, as shown at left in Fig. 13. The gradients of summer balance for Sannomado and Komado Glaciers are positive, consistent with those of a typical glacier. On the other hand, the profiles of the Gozenzawa and Kakunezato Glaciers are nearly flat, particularly in 2015-2016, where it is negative for the Kakunezato Glacier. These glaciers differ from the other two by having a debris-covered area in their downstream parts in ablation years, such as the light snow year of 2015-2016. In general, most glaciers have a positive gradient in both summer and winter, the balance increasing with altitude due to the decrease in temperature. Thus, this trend is consistent with the VSGs here for cases with little-to-no debris cover, but not for cases having a debris-covered area downstream in ablation years.

Now consider the winter balance. In contrast to the usual trend for a larger glacier, the winter balance profiles of these four VSGs are roughly zero or negative. In the light snow year of 2015-2016, the gradient trends of the four glaciers are nearly zero. Even in the heavy snow year of 2016-2017, the gradient of winter balance of the Kakunezato Glacier is nearly zero. Differing even more from a typical glacier, the gradients of winter balance of Gozenzawa, Sannomado, and Komado Glaciers are negative in the heavy snow year. Another common feature of the winter (and summer) balances of these VSG is that their balances are always significantly larger than the average snow depth of the Murododaira (2.93 m weq: Iida et al., 2018). The large contribution to the total balance comes from avalanches and snowdrift and thus likely is the cause of the near-zero and negative profiles. Moreover, compared to the summer balance profiles, those in winter have much greater year-to-year variation.

On the right side of Fig. 13, we show profiles of the annual balance. The annual balance is positive for these VSGs only in the heavy snow year. However, the gradients vary significantly by year and by glacier. For the Sannomado and Komado



Glaciers, the gradients are positive in the light snow year of 2015-2016 and negative in the heavy snow year of 2016-2017. For the Gozenzawa Glacier, the gradient is slightly negative in all years, whereas the gradient for the Kakunezato Glacier is negative in the light snow year and positive in the heavy snow year. Thus, the gradients of the annual balance of the four VSGs differ from the positive trend for typical glaciers and show significant variation between themselves.

**Figure 13 Profiles of the mass balances of Gozenzawa, Sannomado, Komado, Kakunezato Glaciers. The green line is the average snow amount of the Murododaira (2.93 m weq: Iida et al., 2018).**



## 5 Discussion

### 5.1 Mass balance characteristics of VSGs in Japan: comparison to other glaciers

We reported on our measurements of the winter and summer balances and continuous annual balances of the recently classified
VSGs in Japan. To help understand their behaviour, we compared them with previously reported measurements of other glaciers worldwide. In general, the glacier mass balance amplitude increased from polar, to continental, and then to maritime climates. An exception to this trend was the Ural glaciers, which appeared to have anomalously high amplitude for their dry, cold climate. The Urals have small glaciers that inhabit valley bottoms, where the effects of climate and topography are localized, leading to higher mass balance amplitudes (Dyurgerov and Meier, 1999). Like the VSGs studied here, the glaciers
in the Urals lie in narrow valleys, gaining much of their mass through avalanches or snowdrifts. However, while the Urals are dry, the northern Japanese Alps has some of the heaviest snowfall in the world. This combination of heavy snowfall and narrow valleys may explain why the mass balance amplitude of VSGs in Japan was found to be higher than the other glaciers worldwide (Fig. 11). In addition, Braithwaite and Hughes (2020) showed that the mass balance amplitude correlated with summer temperature, which is also relatively high in the northern Japanese Alps. In addition, we found the standard deviation
of the annual balance to be nearly proportional to the mass balance amplitude (Fig. 11), with the result that VSGs in Japan have a variation in annual balance higher than the other glaciers worldwide.

Why do the glaciers exist here? The high winter balance is likely a major factor. We found the winter balance of VSGs in Japan to be the highest of all investigated glaciers worldwide to date, being more than double those of the Kamchatka glaciers, the second highest. However, despite the very heavy snowfall here, the winter snowfall cannot keep pace with the
rapid snowmelt in summer, and hence the perennial snow patches form only where avalanches and snowdrifts deposit significant snow (Higuchi, 1968). Here, the VSGs receive more than double the snowfall amounts due to recharge from avalanches and snowdrifts. This source of snow is thus the reason VSGs can exist in such a warm climate.

Concerning the yearly fluctuations in mass balance, the summer balances of the VSGs here are relatively constant, whereas the winter and annual balances fluctuate significantly from year to year (Fig. 9). This result indicates that the variable
winter balance dominates the behaviour of the annual balance of VSGs in these mountains. Higher mass balance amplitudes result in the annual balance having lower sensitivity to the summer balance and higher sensitivity to the winter balance (Dyurgerov and Meier, 1999).

The profiles of these mass balances were found to differ from those typical of glaciers elsewhere. A typical glacier consists of an upstream accumulation area and a downstream ablation area, separated by an equilibrium line altitude (ELA), a
structure that produces a positive gradient in the annual balance. Even if the annual balance can become an ablation-area throughout due to recent climate change effects (e.g., the Ålfotbreen Glacier (Bjarne Kjøllmoen (Ed.) et al., 2019)), the glacier's gradient can remain positive (Oerlemans and Hoogendoorn, 1989). In these cases, the ELA can be defined above the glacier altitude range.



In contrast, we found that the gradients of mass balance of VSGs in Japan vary greatly in light or heavy snow years, often being near zero or negative. Although the Sannomado and Komado Glaciers had positive summer gradients, the Gozenzawa and Kakunezato Glaciers, where debris-covered areas appear in the downstream parts of the glacier in light snow years, did not. Suppression of melting in the downstream parts in these cases is likely due to their debris cover (e.g., Nicholson and Benn, 2006). In addition, the gradients of winter balance of VSGs in Japan did not have a positive gradient. We argued

that this property of the VSGs is likely due to the significant effect of avalanches and snowdrift, which can be greater downstream. As a result, the gradient of the annual balance of VSGs in Japan can be negative. Furthermore, the annual balances were negative (ablation-area throughout) most years, being positive (accumulation-area throughout) for all VSGs only in the heavy snow year. Taken together, these results suggest that VSGs in Japan are not divided by a distinct glacier ELA into an upstream accumulation zone and a downstream ablation zone.

**5.2 Climate sensitivity of VSGs in Japan**

The VSGs in Japan persist where the winter balance is more than double the snowfall amount due to avalanche accumulation. Thus, of the two types of VSGs (Kuhn 1995), the VSGs in Japan can be classified as topographically controlled VSGs.

        Generally, VSGs are more sensitive and react faster to climatic change than larger glaciers (Hoelzle et al., 2003; Hoffman et al., 2007; Jóhannesson et al., 1989). However, the responses of individual VSGs to changes in climatic forcing

depend on the site, being influenced by topographic factors, feedbacks, and non-linearities (e.g., Carturan et al., 2013; Kuhn, 1995; López-Moreno et al., 2006). Like the VSGs here, other topographically controlled VSGs tend to be strongly correlated with the winter balance and weakly correlated with the summer balance (De Marco et al., 2020; Hughes, 2009; Huss and Fischer, 2016; Pecci et al., 2008). Huss and Fischer (2016), modeling the climate sensitivity of 1,133 VSGs in the Swiss Alps, argued that about 80% of the VSGs in the Swiss Alps would disappear by 2030. However, they reported that the

topographically controlled VSGs are less sensitive to temperature fluctuations and thus some may survive future warming. The lower sensitivity of topographically controlled VSGs to temperature rise has also been reported in the Eastern Alps (Carrivick et al., 2015) and Canadian Rockies (DeBeer and Sharp, 2009).

        To help understand how these topographically controlled VSGs in Japan may respond to climate, we consider their high sensitivity to the winter balance. Their dependence on winter balance suggests a dependence on snowfall depth variation.

How has snowfall depth changed over recent years? Yamaguchi et al. (2011) made meteorological observations in the alpine zone of central Japan, finding no decreasing trend in snow depth in the alpine zone for 1990–2010, yet large yearly fluctuations. Suzuki and Sasaki (2019) found similar results for 2002–2017. Regional climate projections using a high-resolution non-hydrostatic regional climate model (NHRCM) with 5-km resolution and 1-km grid spacing concluded that the amount of snow in the northern Japanese Alps will decrease if the temperature rises by 2 K (Kawase et al., 2020).

If the non-decreasing trend in snow depth continues, VSGs in Japan should persist like other topographically controlled VSGs worldwide. On the other hand, a temperature rise and snow-depth decrease would deplete these VSGs, which





are already less than 50 m thick. If glacier shrinkage continues at the same rate as that in 2015–2019, the VSGs will likely transform into perennial snow patches and then vanish.

Understanding the variation in the annual balance of a glacier requires measurements of the seasonal balance for at least 30 years (Ohmura, 2010). As this study covered only four-years, further mass balance observations of the VSGs in Japan are needed.

## 6 Conclusion

In this study, we calculated the annual balances and seasonal mass balances of VSGs in the northern Japanese Alps using a geodetic method based on SfM–MVS technology and aerial images. Their summer balances showed almost no yearly
fluctuation, whereas they had large yearly fluctuations in the winter balance and annual balance. Therefore, their annual balance was dominated by their winter balance. The annual balance of the entire area represented an accumulation area during a heavy snow year and an ablation area during a lighter snow year. These characteristics differ from those of the typical glacier, which instead have both an upstream accumulation area and a downstream ablation area, separated by a glacier ELA. Moreover, VSGs in Japan had negative annual balance gradients, which suggests that these VSGs did not have an equilibrium line during
the observation period.  A comparison of the mass balance amplitudes of VSGs in Japan with those of other glaciers worldwide showed that mass balance amplitude increased from polar to continental to maritime climates. The VSGs studied here had the highest mass balance amplitudes, probably due to the warm climate with very heavy snowfall. We argued that their continued existence likely depends on their acquiring most of their ice from avalanches. For this reason, we classified them as topographically controlled VSGs.

Like other topographically controlled VSGs worldwide, the VSGs in Japan showed relatively low sensitivity to summer balance fluctuation. However, if the amount of snow in Japan were to decrease due to climate change, these VSGs would likely respond within 10–20 years by significantly shrinking, transforming into perennial snow patches, and then vanishing. Before they experience such climate changes, they should be continually observed for at least 30 years to better predict their future development.




## Appendices

**Appendice A  Area, average winter (ABw) and summer (ABs) mass balance amplitudes (Aα), standard deviations of annual mass balance (sBn), and observation periods of VSGs and Hamaguri-yuki perennial snow patch in the northern Japanese Alps and glaciers around the world.**

| NO. | Name | Region | Lat (°) | Log (°) | Area (km²) | Abw (m weq) | ABs (m weq) | Aα (m weq) | SBn (m weq) | observation periods (Year) |
|---|---|---|---|---|---|---|---|---|---|---|
| 1 | KAKUNEZATO | Northern Japanese Alps | 36.63 | 137.76 | 0.08 | 10390.33 | -10696.34 | 10696.34 | 3438.07 | 2015-2019(4years) |
| 2 | SANNOMADO | Northern Japanese Alps | 36.63 | 137.64 | 0.10 | 9898.99 | -10904.30 | 10401.65 | 3514.50 | 2015-2019(4years) |
| 3 | KOMADO | Northern Japanese Alps | 36.63 | 137.63 | 0.11 | 8900.15 | -9979.81 | 9439.98 | 3364.74 | 2015-2019(4years) |
| 4 | GOZENZAWA | Northern Japanese Alps | 36.57 | 137.62 | 0.03 | 7348.55 | -7847.43 | 7905.34 | 2098.18 | 2015-2019(4years) |
| 5 | KORYTO | Kamchatka | 54.83 | 161.80 | 7.54 | 4430.00 | -4320.80 | 4375.40 | 1696.12 | 1982-2000(5years) |
| 6 | IVORY | New Zealand | -43.13 | 170.93 | 0.80 | 2630.00 | -5013.33 | 3821.67 | 1092.86 | 1970-1975(6years) |
| 7 | HANSEBREEN | Scandinavia | 61.75 | 5.68 | 2.75 | 3426.28 | -4034.72 | 3730.50 | 1511.49 | 1986-2017(32years) |
| 8 | LUPO | Alps | 46.08 | 9.99 | 0.20 | 3354.11 | -3948.44 | 3651.28 | 1108.90 | 2010-2018(9years) |
| 9 | AALFOTBREEN | Scandinavia | 61.75 | 5.65 | 3.98 | 3606.65 | -3692.61 | 3649.63 | 1436.59 | 1963-2017(54years) |
| 10 | OSSOUE | Alps | 42.77 | -0.14 | No record | 2798.13 | -4281.88 | 3540.00 | 1056.16 | 2002-2017(16years) |
| 11 | KOZELSKIY | Kamchatka | 53.23 | 158.82 | 1.80 | 3451.96 | -3608.26 | 3530.11 | 1017.90 | 1973-1997(23years) |
| 12 | NOISY CREEK | Western north America | 48.67 | -121.53 | 0.46 | 3130.87 | -3652.67 | 3391.77 | 1071.96 | 1993-2016(15years) |
| 13 | NORTH KLAWATTI | Western north America | 48.57 | -121.09 | 1.48 | 2904.87 | -3570.47 | 3237.67 | 937.77 | 1993-2016(16years) |
| 14 | HOEGTUVBREEN | Scandinavia | 66.45 | 13.64 | 2.56 | 3210.00 | -3206.14 | 3208.07 | 877.41 | 1971-1977(7years) |
| 15 | SPERRY | Western north America | 48.63 | -113.75 | 0.78 | 2958.46 | -3405.38 | 3181.92 | 844.93 | 2005-2017(13years) |
| 16 | NISQUALLY | Western north America | 46.82 | -121.74 | 6.67 | 2796.00 | -3551.00 | 3173.50 | 670.11 | 2006-2015(10years) |
| 17 | SENTINEL | Western north America | 49.89 | -122.98 | 1.74 | 3180.52 | -3042.39 | 3111.46 | 922.16 | 1966-1989(23years) |
| 18 | SOUTH CASCADE | Western north America | 48.35 | -121.06 | 1.90 | 2790.23 | -3380.04 | 3085.13 | 999.97 | 1959-2017(57years) |
| 19 | SVELGJABREEN | Scandinavia | 59.95 | 6.28 | 22.34 | 3060.73 | -2942.82 | 3001.77 | 1114.09 | 2007-2017(11years) |
| 20 | SURETTA MERID. | Alps | 46.51 | 9.36 | 0.13 | 2358.43 | -3614.14 | 2986.29 | 909.68 | 2010-2018(7years) |
| 21 | CALDERONE | Alps | 42.47 | 13.57 | 0.03 | 2903.75 | -2999.50 | 2951.63 | 1187.81 | 2006-2017(10years) |
| 22 | ENGABREEN | Scandinavia | 66.65 | 13.85 | 36.25 | 3044.50 | -2848.00 | 2946.25 | 1007.17 | 1970-2017(48years) |
| 23 | BLOMSTOELSKARDSBREEN | Scandinavia | 59.95 | 6.33 | 22.54 | 3171.09 | -2657.64 | 2914.36 | 1141.75 | 2007-2017(11years) |
| 24 | SANDALEE | Western north America | 48.41 | -120.79 | 0.18 | 2877.46 | -2917.38 | 2897.42 | 786.63 | 1995-2016(13years) |
| 25 | SVARTISHEIBREEN | Scandinavia | 66.55 | 13.76 | 5.53 | 3053.29 | -2687.71 | 2870.50 | 1114.62 | 1988-1994(7years) |
| 26 | JOSTEFONN | Scandinavia | 61.42 | 6.55 | 3.88 | 2749.60 | -2913.40 | 2831.50 | 924.08 | 1996-2000(5years) |
| 27 | ROLLESTON | New Zealand | -42.89 | 171.53 | 0.11 | 2538.63 | -3075.63 | 2807.13 | 1203.26 | 2011-2018(8years) |
| 28 | OBRUCHEV | Urals | 67.64 | 65.78 | 0.30 | 2666.82 | -2864.55 | 2765.68 | 838.37 | 1958-1979(23years) |
| 29 | ECHAURREN NORTE | Andes | -33.58 | -70.13 | 0.40 | 2473.31 | -2948.48 | 2710.89 | 1548.04 | 1976-2017(42years) |
| 30 | EMMONS | Western north America | 46.85 | -121.72 | 11.27 | 2366.36 | -3029.09 | 2697.73 | 803.34 | 2006-2016(11years) |
| 31 | WOOLSEY | Western north America | 51.11 | -118.06 | 3.92 | 2588.00 | -2772.22 | 2680.11 | 779.98 | 1966-1975(10years) |
| 32 | BREIDABLIKKBREA | Scandinavia | 60.07 | 6.37 | 3.21 | 2298.75 | -3004.38 | 2651.56 | 1274.18 | 1963-2012(16years) |
| 33 | BONDHUSBREA | Scandinavia | 60.03 | 6.33 | 10.67 | 2556.00 | -2682.60 | 2619.30 | 937.54 | 1977-1981(5years) |
| 34 | BREWSTER | New Zealand | -44.07 | 169.43 | 2.03 | 2466.79 | -2749.50 | 2608.14 | 1086.89 | 2005-2018(14years) |
| 35 | TIEDEMANN | Western north America | 51.33 | -125.05 | 62.69 | 1926.67 | -3275.00 | 2600.83 | 168.93 | 1981-1990(6years) |
| 36 | DJANKUAT | Caucasus | 43.19 | 42.76 | 2.69 | 2470.63 | -2719.38 | 2595.00 | 715.47 | 1968-2018(48years) |
| 37 | GRAAFJELLSBREA | Scandinavia | 60.08 | 6.40 | 8.05 | 2356.65 | -2792.41 | 2574.53 | 1302.32 | 1964-2012(17years) |
| 38 | LANGFJORDJOEKELEN | Scandinavia | 70.13 | 21.74 | 3.22 | 2060.85 | -3009.85 | 2535.35 | 734.32 | 1989-2017(27years) |
| 39 | TROLLBERGDALSBREEN | Scandinavia | 66.72 | 14.44 | 1.80 | 2308.60 | -2750.50 | 2529.55 | 831.46 | 1970-1994(10years) |
| 40 | HALLSTAETTER G. | Alps | 47.48 | 13.62 | 2.83 | 1975.33 | -3034.58 | 2504.96 | 644.54 | 2001-2017(12years) |
| 41 | WOLVERINE | Western north America | 60.42 | -148.90 | No record | 2297.48 | -2638.00 | 2467.74 | 1126.28 | 1966-2017(50years) |
| 42 | STORGLOMBREEN | Scandinavia | 66.67 | 14.00 | 61.12 | 2153.80 | -2707.80 | 2430.80 | 1010.57 | 1985-2005(10years) |
| 43 | LANGJOKULL ICE CAP | Iceland | 64.67 | -20.10 | 871.00 | 1774.79 | -3078.86 | 2426.82 | 971.37 | 2001-2017(14years) |
| 44 | IGAN | Urals | 67.58 | 66.00 | 0.81 | 2319.55 | -2519.64 | 2419.59 | 685.54 | 1958-1979(22years) |
| 45 | BLAABREEN | Scandinavia | 60.09 | 6.44 | 2.31 | 2025.33 | -2811.67 | 2418.50 | 1186.35 | 1963-1968(6years) |
| 46 | AUSTDALSBREEN | Scandinavia | 61.82 | 7.35 | 10.63 | 2176.37 | -2616.63 | 2396.50 | 1094.00 | 1988-2017(30years) |
| 47 | RUKLEBREEN | Scandinavia | 60.07 | 6.43 | 1.85 | 2485.00 | -2274.00 | 2379.50 | 993.33 | 1964-1968(5years) |
| 48 | SILVER | Western north America | 48.98 | -121.24 | 0.40 | 2318.07 | -2435.87 | 2376.97 | 796.60 | 1993-2016(15years) |
| 49 | MALADETA | Alps | 42.65 | 0.64 | 0.23 | 2051.04 | -2700.70 | 2375.87 | 751.72 | 1992-2016(23years) |
| 50 | HELM | Western north America | 49.96 | -122.99 | 1.41 | 1893.33 | -2853.73 | 2373.53 | 807.67 | 1976-2017(15years) |





| NO. | Name | Region | Lat (°) | Log (°) | Area (km²) | Abw (m weq) | ABs (m weq) | Aα (m weq) | SBn (m weq) | observation periods (Year) |
|---|---|---|---|---|---|---|---|---|---|---|
| 51 | SCHWARZBACH | Alps | 46.60 | 8.61 | 0.04 | 1936.80 | -2769.00 | 2352.90 | 720.78 | 2013-2017(5years) |
| 52 | BUBA | Caucasus | 42.75 | 43.74 | 3.75 | 2154.92 | -2379.23 | 2267.08 | 306.48 | 1968-1980(13years) |
| 53 | TUNSBERGDALSBREEN | Scandinavia | 61.60 | 7.05 | 52.21 | 2218.43 | -2302.57 | 2260.50 | 1197.82 | 1966-1972(7years) |
| 54 | EYJABAKKAJOKULL | Iceland | 64.65 | -15.58 | 112.00 | 1988.40 | -2514.90 | 2251.65 | 809.54 | 1991-2017(20years) |
| 55 | VESLEDALSBREEN | Scandinavia | 61.85 | 7.26 | 4.10 | 2040.17 | -2429.17 | 2234.67 | 1120.75 | 1967-1972(6years) |
| 56 | SARENNES | Alps | 45.12 | 6.13 | 0.10 | 1637.47 | -2772.57 | 2205.02 | 1214.02 | 1949-2018(70years) |
| 57 | OKSTINDBREEN | Scandinavia | 66.02 | 14.29 | 14.01 | 2294.45 | -2111.30 | 2202.88 | 980.06 | 1987-1997(11years) |
| 58 | BENCH | Western north America | 51.43 | -124.92 | 10.35 | 1863.75 | -2523.75 | 2193.75 | 400.50 | 1981-1990(8years) |
| 59 | RUNDVASSBREEN | Scandinavia | 67.30 | 16.06 | 10.85 | 1846.90 | -2533.50 | 2190.20 | 947.48 | 2002-2017(10years) |
| 60 | THRANDARJOKULL | Iceland | 64.70 | -14.88 | 19.40 | 2086.67 | -2268.33 | 2177.50 | 787.75 | 1991-1996(6years) |
| 61 | NIGARDSBREEN | Scandinavia | 61.72 | 7.13 | 46.61 | 2227.86 | -2113.48 | 2170.67 | 965.51 | 1962-2017(56years) |
| 62 | TSANFLEURON | Alps | 46.32 | 7.23 | 2.47 | 1467.38 | -2869.13 | 2168.25 | 991.74 | 2010-2017(8years) |
| 63 | PLACE | Western north America | 50.43 | -122.60 | 3.45 | 1710.44 | -2597.68 | 2154.06 | 762.04 | 1965-2017(25years) |
| 64 | GOLDBERG K. | Alps | 47.04 | 12.97 | 1.03 | 1687.19 | -2558.50 | 2122.84 | 669.49 | 2001-2017(16years) |
| 65 | BRIDGE | Western north America | 50.82 | -123.57 | 88.10 | 1864.00 | -2334.00 | 2099.00 | 787.93 | 1981-1985(5years) |
| 66 | REMBESDALSKAAKA | Scandinavia | 60.54 | 7.37 | 17.26 | 2056.25 | -2136.36 | 2096.31 | 1088.30 | 1963-2017(55years) |
| 67 | SEX ROUGE | Alps | 46.33 | 7.22 | 0.26 | 1455.33 | -2705.67 | 2080.50 | 950.81 | 2012-2017(6years) |
| 68 | TUNGNAARJOKULL | Iceland | 64.32 | -18.07 | 345.00 | 1569.72 | -2587.78 | 2078.75 | 927.48 | 1986-2017(18years) |
| 69 | PENDENTE (VEDR.) / HANGENDERF. | Alps | 46.97 | 11.22 | 0.85 | 1463.86 | -2692.71 | 2078.29 | 666.57 | 1999-2017(14years) |
| 70 | OMNSBREEN | Scandinavia | 60.63 | 7.48 | 1.50 | 1612.00 | -2536.00 | 2074.00 | 1190.31 | 1966-1970(5years) |
| 71 | ANDREI | Western north America | 56.93 | -130.97 | 91.89 | 1810.00 | -2263.33 | 2036.67 | 396.67 | 1978-1990(9years) |
| 72 | ALEXANDER | Western north America | 57.10 | -130.82 | 5.74 | 1676.25 | -2391.25 | 2033.75 | 522.52 | 1978-1990(8years) |
| 73 | PLAINE MORTE | Alps | 46.38 | 7.49 | 7.41 | 1325.63 | -2721.63 | 2023.63 | 813.03 | 2010-2017(8years) |
| 74 | PIZOL | Alps | 46.96 | 9.39 | 0.06 | 1401.00 | -2639.00 | 2020.00 | 482.25 | 2007-2017(11years) |
| 75 | BASODINO | Alps | 46.42 | 8.48 | 1.76 | 1704.44 | -2331.28 | 2017.86 | 776.10 | 1993-2017(25years) |
| 76 | HOFSJOKULL E | Iceland | 64.80 | -18.58 | 213.10 | 1651.38 | -2254.83 | 1953.10 | 880.68 | 1989-2017(29years) |
| 77 | HARBARDSBREEN | Scandinavia | 61.70 | 7.68 | 13.17 | 1780.40 | -2099.00 | 1939.70 | 673.83 | 1997-2001(5years) |
| 78 | SANKT ANNA | Alps | 46.60 | 8.60 | 0.18 | 1525.00 | -2349.33 | 1937.17 | 442.34 | 2012-2017(6years) |
| 79 | EKLUTNA EAST BRANCH | Western north America | 61.23 | -148.97 | No record | 1637.50 | -2225.00 | 1931.25 | 843.86 | 2008-2015(8years) |
| 80 | FORNO | Alps | 46.30 | 9.70 | 8.95 | 1609.00 | -2247.17 | 1928.08 | 283.29 | 1955-1960(6years) |
| 81 | SYKORA | Western north America | 50.87 | -123.58 | 25.35 | 1858.00 | -1998.00 | 1928.00 | 752.85 | 1976-1985(10years) |
| 82 | HOFSJOKULL SW | Iceland | 64.72 | -19.05 | 48.80 | 1631.67 | -2142.50 | 1887.08 | 1145.15 | 2006-2017(12years) |
| 83 | BREIDAMJOKULL E. B. | Iceland | 64.22 | -16.33 | 995.00 | 1295.20 | -2414.80 | 1855.00 | 315.31 | 2001-2005(5years) |
| 84 | BLOMSTERSKARDSBREEN | Scandinavia | 68.34 | 17.85 | 2.18 | 1772.67 | -1922.33 | 1847.50 | 772.04 | 2007-2017(11years) |
| 85 | EKLUTNA | Western north America | 61.25 | -148.99 | 31.60 | 1622.27 | -2058.64 | 1840.45 | 738.67 | 1986-2015(11years) |
| 86 | WURTEN K. | Alps | 47.03 | 13.00 | 0.34 | 1403.21 | -2258.72 | 1830.97 | 520.48 | 1983-2011(29years) |
| 87 | ZAVISHA | Western north America | 50.79 | -123.41 | 6.49 | 1750.00 | -1888.00 | 1819.00 | 680.83 | 1976-1985(15years) |
| 88 | HINTEREIS F. | Alps | 46.80 | 10.77 | 6.39 | 1186.33 | -2413.67 | 1800.00 | 754.18 | 2013-2018(6years) |
| 89 | CARESER ORIENTALE | Alps | 46.45 | 10.70 | 1.43 | 881.33 | -2689.67 | 1785.50 | 673.33 | 2006-2011(6years) |
| 90 | EKLUTNA WEST BRANCH | Western north America | 61.23 | -149.01 | No record | 1475.00 | -2063.75 | 1769.38 | 763.31 | 2008-2015(8years) |
| 91 | SATUJOKULL | Iceland | 64.92 | -18.83 | 90.60 | 1623.00 | -1905.00 | 1764.00 | 864.37 | 1988-1997(10years) |
| 92 | CARESER OCCIDENTALE | Alps | 46.45 | 10.69 | 0.19 | 998.50 | -2526.83 | 1762.67 | 669.56 | 2006-2011(6years) |
| 93 | DE LOS TRES | Andes | -49.33 | -73.00 | 0.81 | 1575.29 | -1931.29 | 1753.29 | 816.35 | 1996-2018(7years) |
| 94 | BRUARJOKULL | Iceland | 64.67 | -16.17 | 1525.00 | 1652.60 | -1849.60 | 1751.10 | 781.48 | 1993-2017(15years) |
| 95 | YURI | Western north America | 56.98 | -130.68 | 3.58 | 1435.00 | -2047.78 | 1741.39 | 610.58 | 1978-1990(10years) |
| 96 | TARFALAGLACIAEREN | Scandinavia | 67.93 | 18.64 | 1.01 | 1647.70 | -1834.55 | 1741.13 | 1090.38 | 1986-2015(20years) |
| 97 | HOFSJOKULL N | Iceland | 64.95 | -18.92 | 73.70 | 1456.67 | -2015.33 | 1736.00 | 674.11 | 1988-2017(15years) |
| 98 | KLEINFLEISS K. | Alps | 47.05 | 12.95 | 0.79 | 1420.06 | -2051.63 | 1735.84 | 658.00 | 2001-2017(16years) |
| 99 | KOLDUKVISLARJ. | Iceland | 64.58 | -17.83 | 300.00 | 1482.00 | -1988.82 | 1735.41 | 848.55 | 1992-2017(17years) |
| 100 | GRIES | Alps | 46.44 | 8.34 | 4.35 | 1322.98 | -2121.77 | 1722.38 | 846.70 | 1962-2018(57years) |



| NO. | Name | Region | Lat (°) | Log (°) | Area (km²) | Abw (m weq) | ABs (m weq) | Aα (m weq) | SBn (m weq) | observation periods (Year) |
|---|---|---|---|---|---|---|---|---|---|---|
| 101 | KARSOJIETNA | Scandinavia | 68.36 | 18.32 | 1.23 | 1887.50 | -1513.33 | 1700.42 | 351.82 | 1982-1993(8years) |
| 102 | MALAVALLE (VEDR. DI) / UEBELTALF. | Alps | 46.95 | 11.19 | 6.03 | 1243.23 | -2118.54 | 1680.88 | 485.17 | 2005-2017(13years) |
| 103 | JAMTAL F. | Alps | 46.86 | 10.16 | 2.79 | 1188.04 | -2171.54 | 1679.79 | 640.00 | 1989-2018(26years) |
| 104 | CARESER | Alps | 46.45 | 10.71 | 0.96 | 991.94 | -2350.28 | 1671.11 | 1084.43 | 1976-2018(18years) |
| 105 | ZETTALUNITZ/MULLWITZ K. | Alps | 47.08 | 12.38 | 2.78 | 1221.42 | -2101.75 | 1661.58 | 548.26 | 2007-2018(12years) |
| 106 | ALBIGNA | Alps | 46.30 | 9.64 | 5.68 | 1385.00 | -1906.83 | 1645.92 | 350.12 | 1955-1960(6years) |
| 107 | CIARDONEY | Alps | 45.52 | 7.39 | 0.57 | 1166.35 | -2067.50 | 1616.92 | 871.78 | 1992-2017(26years) |
| 108 | STORBREEN | Scandinavia | 61.57 | 8.13 | 5.14 | 1416.35 | -1770.28 | 1593.31 | 723.70 | 1949-2017(75years) |
| 109 | PLATTALVA | Alps | 46.83 | 8.99 | 0.80 | 1530.08 | -1643.55 | 1586.82 | 644.02 | 1948-1985(38years) |
| 110 | RHONE | Alps | 46.62 | 8.40 | 15.52 | 1358.69 | -1810.80 | 1584.74 | 525.81 | 1885-2017(35years) |
| 111 | MITTIVAKKAT | Greenland | 65.70 | -37.80 | 15.94 | 1203.08 | -1950.00 | 1576.54 | 569.76 | 1996-2015(13years) |
| 112 | KESSJEN | Alps | 46.07 | 7.93 | 0.45 | 1447.45 | -1682.83 | 1565.14 | 692.44 | 1956-1995(40years) |
| 113 | FONTANA BIANCA / WEISSBRUNNF. | Alps | 46.48 | 10.77 | 0.40 | 1116.13 | -2006.23 | 1561.18 | 753.13 | 1984-2017(31years) |
| 114 | OCHSENTALER G. | Alps | 46.85 | 10.10 | 2.59 | 1311.71 | -1795.43 | 1553.57 | 511.23 | 1992-1998(7years) |
| 115 | STORGLACIAEREN | Scandinavia | 67.90 | 18.57 | 2.90 | 1418.90 | -1677.89 | 1548.40 | 701.05 | 1946-2018(73years) |
| 116 | VERMUNT G. | Alps | 46.85 | 10.13 | 2.16 | 1052.25 | -2035.25 | 1543.75 | 527.75 | 1991-1998(8years) |
| 117 | DYNGJUJOKULL | Iceland | 64.67 | -17.00 | 1060.00 | 1615.57 | -1440.86 | 1528.21 | 848.10 | 1992-2017(14years) |
| 118 | RIUKOJIETNA | Scandinavia | 68.08 | 18.05 | 2.65 | 1255.12 | -1798.50 | 1526.81 | 674.81 | 1986-2017(23years) |
| 119 | RIES OCC. (VEDR. DI) / RIESERF. WESTL. | Alps | 46.90 | 12.10 | 1.70 | 1122.89 | -1898.44 | 1510.67 | 684.87 | 2009-2017(9years) |
| 120 | LUNGA (VEDRETTA) / LANGENF. | Alps | 46.47 | 10.62 | 1.60 | 970.46 | -2047.46 | 1508.96 | 657.47 | 2004-2017(13years) |
| 121 | STORSTEINSFJELLBREEN | Scandinavia | 68.22 | 17.92 | 5.91 | 1604.70 | -1379.60 | 1492.15 | 598.45 | 1964-1995(10years) |
| 122 | LIMMERN | Alps | 46.81 | 8.98 | 2.41 | 1385.08 | -1574.63 | 1479.86 | 657.78 | 1948-1985(38years) |
| 123 | GULKANA | Western north America | 63.28 | -145.43 | 31.30 | 1163.22 | -1758.20 | 1460.71 | 643.80 | 1966-2017(50years) |
| 124 | VENEDIGER K. | Alps | 47.13 | 12.34 | 1.99 | 1141.00 | -1779.20 | 1460.10 | 548.09 | 2013-2017(5years) |
| 125 | PEYTO | Western north America | 51.66 | -116.56 | 11.74 | 1129.32 | -1784.29 | 1456.80 | 525.70 | 1966-2017(28years) |
| 126 | CORBASSIERE | Alps | 45.98 | 7.29 | 15.08 | 1034.63 | -1826.68 | 1430.66 | 514.11 | 1997-2017(19years) |
| 127 | SCHWARZBERG | Alps | 46.02 | 7.93 | 5.10 | 1293.25 | -1556.90 | 1425.08 | 797.20 | 1956-2017(60years) |
| 128 | SILVRETTA | Alps | 46.85 | 10.08 | 2.62 | 1236.57 | -1604.22 | 1420.40 | 783.83 | 1919-2018(100years) |
| 129 | AUSTRE MEMURUBREEN | Scandinavia | 61.55 | 8.50 | 8.75 | 1064.40 | -1746.80 | 1405.60 | 547.99 | 1968-1972(5years) |
| 130 | GIETRO | Alps | 46.00 | 7.38 | 5.27 | 1193.71 | -1595.78 | 1394.74 | 664.13 | 1967-2017(49years) |
| 131 | VESTRE MEMURUBREEN | Scandinavia | 61.53 | 8.45 | 9.17 | 1156.80 | -1597.60 | 1377.20 | 565.46 | 1968-1972(5years) |
| 132 | RABOTS GLACIAER | Scandinavia | 67.91 | 18.50 | 3.13 | 1162.09 | -1591.34 | 1376.72 | 632.12 | 1982-2017(32years) |
| 133 | LA MARE (VEDRETTA DE) | Alps | 46.43 | 10.63 | 1.99 | 982.71 | -1745.43 | 1364.07 | 654.26 | 2004-2017(14years) |
| 134 | FINDELEN | Alps | 46.00 | 7.87 | 12.89 | 1083.40 | -1601.60 | 1342.50 | 443.09 | 2005-2017(10years) |
| 135 | GRAASUBREEN | Scandinavia | 61.66 | 8.60 | 2.12 | 1136.83 | -1540.65 | 1338.74 | 622.41 | 1962-2017(56years) |
| 136 | MARMAGLACIAEREN | Scandinavia | 68.08 | 18.68 | 3.31 | 1082.59 | -1528.89 | 1305.74 | 562.78 | 1990-2017(27years) |
| 137 | HELLSTUGUBREEN | Scandinavia | 61.56 | 8.44 | 2.90 | 1095.84 | -1492.84 | 1294.34 | 664.96 | 1962-2017(56years) |
| 138 | GARABASHI | Caucasus | 43.30 | 42.47 | 4.00 | 1106.18 | -1457.12 | 1281.65 | 459.58 | 1984-2018(62years) |
| 139 | CONCONTA NORTE | Andes | -29.98 | -69.65 | 0.10 | 534.83 | -1978.83 | 1256.83 | 756.15 | 2008-2015(6years) |
| 140 | BROWN SUPERIOR | Andes | -29.98 | -69.64 | 0.18 | 547.83 | -1960.17 | 1254.00 | 700.13 | 2008-2015(7years) |
| 141 | GRAND ETRET | Alps | 45.48 | 7.22 | 0.53 | 955.73 | -1429.73 | 1192.73 | 635.90 | 2000-2015(11years) |
| 142 | PILOTO ESTE | Andes | -32.59 | -70.14 | No record | 957.50 | -1419.58 | 1188.54 | 860.47 | 1980-2003(24years) |
| 143 | ALLALIN | Alps | 46.05 | 7.93 | 9.65 | 1094.57 | -1275.15 | 1184.86 | 663.39 | 1956-2017(60years) |
| 144 | MURTEL VADRET DAL | Alps | 46.41 | 9.82 | 0.30 | 935.40 | -1404.80 | 1170.10 | 722.71 | 2013-2017(5years) |
| 145 | HANSBREEN | Svalbard | 77.08 | 15.63 | 56.74 | 960.23 | -1324.46 | 1142.35 | 371.34 | 1989-2017(26years) |
| 146 | VERNAGT F. | Alps | 46.88 | 10.82 | 7.08 | 906.84 | -1356.35 | 1131.60 | 522.87 | 1966-2017(51years) |
| 147 | GROSSER ALETSCH | Alps | 46.50 | 8.03 | 83.02 | 848.27 | -1310.47 | 1079.37 | 615.05 | 1940-1999(60years) |
| 148 | RAM RIVER | Western north America | 51.85 | -116.48 | 1.80 | 891.11 | -1211.11 | 1051.11 | 482.55 | 1966-1975(9years) |
| 149 | QAPIARFIUP SER. | Greenland | 65.58 | -52.21 | 20.85 | 1120.00 | -962.00 | 1041.00 | 602.58 | 1981-1985(5years) |
| 150 | ADLER | Alps | 46.01 | 7.87 | 1.98 | 823.44 | -1210.67 | 1017.06 | 423.53 | 2009-2017(12years) |





| NO. | Name | Region | Lat (°) | Log (°) | Area (km²) | Abw (m weq) | ABs (m weq) | Aα (m weq) | SBn (m weq) | observation periods (Year) |
|---|---|---|---|---|---|---|---|---|---|---|
| 151 | MARTIAL ESTE | Andes | -54.78 | -68.40 | 0.09 | 934.75 | -1041.00 | 987.88 | 576.28 | 2001-2017(16years) |
| 152 | AUSTRE LOVENBREEN | Svalbard | 78.87 | 12.15 | No record | 688.64 | -1196.55 | 942.59 | 420.88 | 2008-2018(11years) |
| 153 | VOERINGBREEN | Svalbard | 78.04 | 13.95 | No record | 604.33 | -1258.67 | 931.50 | 392.78 | 1974-1988(15years) |
| 154 | TS.TUYUKSUYSKIY | HMA (high mountain Asia) | 43.05 | 77.08 | 2.26 | 696.87 | -1147.27 | 922.07 | 505.29 | 1965-2017(52years) |
| 155 | AUSTRE BROEGGERBREEN | Svalbard | 78.89 | 11.83 | 6.12 | 641.13 | -1168.15 | 904.64 | 374.26 | 1967-2017(40years) |
| 156 | FREYA | Greenland | 74.38 | -20.82 | 5.30 | 748.29 | -1059.86 | 904.07 | 581.18 | 2008-2017(7years) |
| 157 | MIDTRE LOVENBREEN | Svalbard | 78.88 | 12.05 | 5.45 | 697.63 | -1105.90 | 901.76 | 338.53 | 1976-2017(40years) |
| 158 | LEVIY AKTRU | HMA (high mountain Asia) | 50.08 | 87.69 | 5.95 | 835.22 | -954.35 | 894.78 | 355.01 | 1977-200(23years) |
| 159 | KARA-BATKAK | HMA (high mountain Asia) | 42.14 | 78.27 | 2.47 | 504.70 | -1244.91 | 874.80 | 336.24 | 1976-2018(23years) |
| 160 | MALIY AKTRU | HMA (high mountain Asia) | 50.05 | 87.75 | 2.73 | 789.95 | -874.20 | 832.08 | 435.53 | 1962-2009(44years) |
| 161 | BELLINGSHAUSEN | Antarctic | -62.17 | -58.89 | 11.02 | 750.60 | -900.20 | 825.40 | 422.21 | 2008-2012(5years) |
| 162 | SHUMSKIY | HMA (high mountain Asia) | 45.08 | 80.23 | 2.81 | 442.12 | -1129.33 | 785.73 | 456.05 | 1967-1991(26years) |
| 163 | WALDEMARBREEN | Svalbard | 78.68 | 12.07 | 2.50 | 496.93 | -1069.07 | 783.00 | 239.37 | 1996-2009(14years) |
| 164 | KONGSVEGEN | Svalbard | 78.80 | 12.98 | 101.90 | 691.97 | -777.53 | 734.75 | 368.91 | 1987-2017(30years) |
| 165 | GOLUBIN | HMA (high mountain Asia) | 42.46 | 74.50 | 5.45 | 719.85 | -720.91 | 720.38 | 339.97 | 1969-2018(34years) |
| 166 | JOHNSONS | Antarctic | -62.67 | -60.35 | 5.36 | 785.00 | -591.25 | 688.13 | 339.09 | 2002-2017(16years) |
| 167 | HURD | Antarctic | -62.69 | -60.40 | 4.03 | 642.50 | -685.63 | 664.06 | 423.02 | 2002-2017(16years) |
| 168 | SARY TOR (NO.356) | HMA (high mountain Asia) | 41.83 | 78.17 | 2.64 | 424.78 | -903.33 | 664.06 | 470.94 | 1985-2018(9years) |
| 169 | VALHALTINDE | Greenland | 61.44 | -45.32 | 1.90 | 496.00 | -662.00 | 579.00 | 256.18 | 1979-1983(5years) |
| 170 | BATYSH SOOK/SYEK ZAPADNIY | HMA (high mountain Asia) | 41.79 | 77.75 | 1.01 | 242.73 | -863.82 | 553.27 | 300.64 | 1971-2018(11years) |
| 171 | KRONEBREEN | Svalbard | 78.97 | 13.18 | No record | 500.00 | -606.00 | 553.00 | 201.54 | 2013-2017(5years) |
| 172 | AMITSULOQ ICE CAP | Greenland | 66.14 | -50.32 | 198.00 | 494.33 | -562.22 | 528.28 | 305.06 | 1982-1990(9years) |
| 173 | NORDENSKIOELDBREEN | Svalbard | 78.70 | 17.18 | No record | 486.77 | -552.31 | 519.54 | 256.52 | 2006-2018(13years) |
| 174 | GLACIER NO. 354 (AKSHIYRAK) | HMA (high mountain Asia) | 41.80 | 78.15 | 6.38 | 251.38 | -780.13 | 515.75 | 238.26 | 2011--2018(8years) |
| 175 | URUMQI GLACIER NO. 1 E-BRANCH | HMA (high mountain Asia) | 43.11 | 86.81 | 1.02 | 124.19 | -822.94 | 473.56 | 395.16 | 1996-2017(16years) |
| 176 | URUMQI GLACIER NO. 1 W-BRANCH | HMA (high mountain Asia) | 43.12 | 86.80 | 0.57 | 138.70 | -772.10 | 455.40 | 394.51 | 1996-2017(20years) |
| 177 | URUMQI GLACIER NO. 1 | HMA (high mountain Asia) | 43.12 | 86.81 | 1.59 | 172.79 | -672.25 | 422.52 | 393.55 | 1989-2017(24years) |
| 178 | MELVILLE SOUTH ICE CAP | Arctic north America | 75.40 | -115.00 | 51.00 | 185.63 | -540.58 | 363.10 | 433.28 | 1963-2017(43years) |
| 179 | MEIGHEN ICE CAP | Arctic north America | 79.95 | -99.13 | 58.00 | 171.92 | -370.14 | 271.03 | 375.52 | 1960-2017(51years) |
| 180 | CHACALTAYA | Andes | -16.35 | -68.13 | 0.03 | 17.60 | -484.00 | 250.80 | 709.83 | 2000-2004(5years) |
| 181 | QIYI | HMA (high mountain Asia) | 39.24 | 97.76 | 2.98 | 316.80 | -124.80 | 220.80 | 185.52 | 1975-1985(5years) |
| 182 | DEVON ICE CAP NW | Arctic north America | 75.42 | -83.25 | 1688.00 | 115.43 | -264.52 | 189.97 | 195.27 | 1961-2017(56years) |
| 183 | COMMONWEALTH | Dry Valley | -77.55 | 163.08 | 52.20 | 9.47 | -19.26 | 14.37 | 36.75 | 1994-2012(23years) |
| 184 | HOWARD | Dry Valley | -77.76 | 163.51 | 8.30 | -2.13 | -8.39 | 5.26 | 32.25 | 1994-2012(19years) |

**Code and data availability**

The full sample of glaciological observations for individual glaciers is publicly available from the WGMS (doi:10.5904/WGMS-FOG-2020-08, 2020; WGMS, 2020).

**Author contributions**

Authors K.A., C.N. and R.Y. conducted the field survey and performed an analysis of field data. K.A. wrote the paper. C.N.
and C.N., K.F. and H.I. checked and improved the manuscript and suggested some discussion points. All authors have read and agreed to the published version of the manuscript.



**Competing interests**

The authors declare that they have no conflict of interest.

**Acknowledgements**

We would like to express thanks to the captain of Cessna aircraft: T. Kinoshita (New Central Airservice Co.), K. Tomooka (IBEX Aviation Co.), and H. Miyata (Tokyo Koku Co.), Mountain Guides: S. Arakawa, S. Matsumoto, and Y. Ito, Niigata University: H. Hitomi, Y. Mori, H, Takadama, N. Sakurai, H. Sugiyama, K. Matsumoto, R. Yoshimura, A. Ayano, and N. Yamada. This study was supported by Toyama Prefecture Energetic Snow Country Creation Project in 2015 and 2016, Research commissioned fee from Hakuba village, Nagano prefecture in 2018, and Research Grant from the Japan landslide

society in 2019. We appreciate the Matsumoto Sabo office, Tateyama mountain area Sabo office, and Kurobe river office of the Ministry of Land, Infrastructure, Transport and Tourism Hokuriku Regional Development Bureau for providing high-resolution aerial laser data.

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
