# Peer review of "Characteristics of mountain glaciers in the northern Japanese Alps"

_The Cryosphere, 2021_

## Author Comment (AC1)

**Dear Anonymous Referee #1 (corresponding to https://doi.org/10.5194/tc-2021-182-RC1)**

We thank you for your valuable comments. Here we address how we revise the manuscript corresponding to your specific comments. The comments are in black words, which are followed by our responses in red words.

**Comment**

This paper describes an interesting and well-conducted study on very small glaciers in the Northern Japanese Alps. Although these tiny snow/ice patches probably have a very limited relevance, their monitoring allows new insights into glaciological processes determining this glacier size class. The evaluation of data specific to these miniature glaciers in comparison with worldwide observations results in valuable conclusions. The paper is mostly clearly written and illustrated and fits well into The Cryosphere. However, when reading the manuscript, a relatively large amount of minor issues and questions came up. At several instances, wrong units and (maybe) wrongly stated results are present that require careful re-reading by the authors. I have two more important conceptual comments that should be addressed during the revision:

**Response**

The following are responses to the substantive and detail comments.

**Substantive comments:**

1:Computation of mass balance profiles: The authors present mass balance profiles and elevation gradients for both winter and annual balance. Although they correctly mention and that the comparison of digital elevation models does not deliver local mass balance (allowing the computation of gradients etc.) and discuss emergence velocities of the ice, the conceptual approach remains partly vague and might be questionable for some situations: In fact, only for one of the five investigated glaciers, emergence velocities have been determined in the field. Moreover, the investigated locations only cover a limited elevation range, about at the median glacier elevation. More justification should be provided for the assumption that the measured emergence velocities are directly transferable to the other glaciers, and that the values measured are representative for the entire glacier. Conceptually, for typical alpine glaciers, emergence velocities are small around the median elevation, but increase in magnitude towards the top and the snout. If this also applies to the present situation on glaciers in the Japanese Alps, it might be that the measurements just captured the low emergence velocities at median elevation but the signal across the entire elevation range has been missed and mass balance profiles derived from surface elevation changes are thus biased. I do not consider this possibility as likely as the measured emergence velocities are much smaller than the mass balance rates, and the dynamics

on the snow/ice patches is certainly different than on a standard glacier. Nevertheless, the issue needs to be looked at more closely, providing more justification for the assumptions.

**Response**

We change a profile of altitude changes instead of a mass balance profile in Figure 13, because accurate estimates of emergence velocities are difficult. In the Changri Nup glacier in Nepal, the emergence velocity of the glacier terminus is about 0.37 m a$^{-1}$ against an average horizontal velocity of 9.7 m a$^{-1}$ (Vincent et al., 2016). In the Argentière Glacier in the French Alps, the emergence velocity of the glacier terminus is 3-6 m a$^{-1}$ against a horizontal velocity of 35-60 m a$^{-1}$ (Vincent et al., 2021). As shown in these references, the emergence velocity is less than a tenth of the horizontal velocity on the glacier surface. In the case of VSGs in the Japanese Alps, the horizontal velocity is less than 4 m a$^{-1}$, and we consider that the emergence velocity is extremely small. In our revised paper, we add the above explanation with some references, to explain that the emergence velocity of VSGs in the Japanese Alps is extremely small. We discuss the characteristics of the mass balance profile using the profile of the altitude change as an estimation that the emergence velocity is low.

2:Converting snow/ice volume changes to mass change: The study is based on surface elevation changes that are subsequently converted to a mass change using a density assumption, both for winter and annual balance. In my opinion, there is a high potential for uncertainty that is barely described in the paper so far. The authors rely on some local observations of snow/firn/ice density and consider these values as universal, both regarding all glaciers and all years. This is certainly too much simplified. A variability in densities in the spatio-temporal domain is certain. It is clear that this cannot be measured but an additional uncertainty component that should be estimated based on as much evidence as possible is clearly needed. Furthermore, I am also partly doubtful regarding the chosen numbers and mentioned processes: (1) The density of winter snow corresponds to the observations at an off-glacier snow observatory. The authors mention that snow depth is half that on the glaciers, most likely resulting in smaller densities. Moreover, I would expect the winter snow depth on the glaciers to be particularly high because of a significant portion of wind drifted and avalanche snow. (2) The density of volume change is actually not only composed of the first annual layer's density, as suggested by the authors, but strongly depends on compaction dynamics in older layers. Although it is argued that transition from snow to ice is occurring during a single year under this climate, more evidence supporting this claim is necessary in my opinion. If incompressible glacier ice (900 kg m-3) is formed during a single year, this should become evident in field observations and images after a year with mass loss, which has not been demonstrated. Figure 2 seems to indicate snow surface almost everywhere. Also, even if compaction of older layers was zero, the strong variability in annual mass balance should lead to differences in the density of volume change between negative (high density of lost material) and positive (lower density of gained material) years – an effect that is not considered at

present. I realize that it will be difficult to resolve all these processes without any further observations (that would be difficult if not impossible to acquire). But the problem should be carefully discussed and be incorporated in a better uncertainty estimate of seasonal/annual mass balances. Uncertainties presently are too small in my opinion.

**Response**

As you pointed out, the winter snow density includes uncertainty. Therefore, the annual mass balance does not coincide with the sum of the winter balance and summer balance. In our revised paper, we exclude the winter and summer balances in Table 4, Figures 9 and 13, and instead shown as accumulation depth and ablation depth. However, to compare with the mass balance amplitude of glaciers which is recorded in WGMS, we need to calculate the winter and summer balances of the northern Japanese Alps. In Japanese mountains, the snow density of avalanche deposits is larger than that of snowfall snow layers ( Naruse et al., 1986; Abe et al., 2016). In the northern Japanese Alps, the density of avalanche deposits is 590 kg/m$^3$ (Shimizu et al., 1974). Therefore, we calculate the mass balance amplitude using the winter snow density in the range of 431 to 590 kg/m$^3$. Figures 11 and 12 and the appendix are also revised accordingly.

The transition from snow to ice occurred during a single year. We already confirmed an exposed ice layer on the glacier surface at the end of the snowmelt season of negative balance year based on aerial images and field (in Fig 4a of Arie et al., 2019). Therefore, we use 695 kg m$^{-3}$ for positive balance years and 860 kg m$^{-3}$ for negative balance years as the snow density. These densities are derived from measurement data by Kawashima (1997) and Fukui et al. (2018).

**Detailed: comments:**

Line 23: Why no equilibrium line? The ELA was just either above or below the glacier.

**Response**

We change to 'VSGs in Japan are not divided by a distinct glacier ELA into an upstream accumulation zone and a downstream ablation zone'

Line 31: only define acronyms if they are also used later in the paper.

**Response**

We remove 'GPR' as we do not use any more.

Line 58: "volume" and not "mass" change results from the geodetic method.

**Response**

We change it.

Line 59: Given the multitude of recent studies relying on the geodetic method for glacier change assessment, I would suggest selecting newer and more appropriate references on the methodology.

**Response**

We add some references.

Line 64: It would make sense to mention already here that the geodetic method does not provide local mass balance, and thus mass balance profiles, but only elevation change. Given an estimate of spatially distributed (!) emergence velocities AND local density, this elevation change can be converted into a mass balance profile (see major comment above).

**Response**

As responded second substantive comment above, in our revised paper, we show it as a profile of relative altitude change.

Figure 1: Inset in map is not very clear. It would help to show entire Japan and use colour/shading for the sea.

**Response**

We change it.

Figure 2: mention the year of the images in the caption.

**Response**

The year of shooting is already listed in 2016.

Table 1: The area stated in the last column is wrong (typo).

**Response**

We change your point in Table 1. For Karamatsuzawa Glacier, the correct value is 0.103 km$^2$.

Line 92: At least somewhere in a table, the exact dates of the measurements need to be given. Considering the high mass balance amplitudes, daily ablation/accumulation rates are important, thus also time differences of a few days will be affecting the stated seasonal/annual mass balances.

**Response**

We add the date of the aerial photography.

Figure 3: Would be easier to read with a larger contour line spacing.

**Response**

Although we created 15m and 20m contour lines, the topography of the surrounding mountains has become difficult to see in detail. We do not change the figure.

Line 135: rho is not the ice density but the average density of the lost or gained material, including the effects of the compaction of lower firn/ice layers.

**Response**

We change to 'snow and ice density'

Line 137: It is not strictly speaking the "stratigraphic system"! This system refers to the absolute maximum (winter) and minimum (autumn) of glacier mass. This date is normally unknown and can rarely exactly be met by monitoring programmes. I am quite sure that the surveys conducted do not correspond to this stratigraphic system (which is not a problem but needs to be stated). Regarding the stratigraphic system, referring to the much newer Cogley et al. 2011 publication would probably be more appropriate.

**Response**

We add Cogley et al. (2011) as a new reference. It is not easy to take aerial images before a snowy day, because Cessna flies on clear fine days. As you pointed out, our flight does not perfectly match the stratigraphic system. In line 138, we add the sentence as "The aerial images were taken just before the snow falls at the end of snowmelt season based on the stratigraphic system. However, the date has a gap of several days from a day before the snow falls, due to flight schedule and weather conditions."

Line 154: The choice of using the smallest glacier area over the study period for inferring total mass balance is interesting and should be discussed in more depth. In fact, reference-surface mass balances

are then computed that may diverge quite a bit from the conventional mass balance, typically reported in glacier monitoring programmes, related to the actual glacier surface area. Both systems have their pro and cons. The mass gain or loss occurring beyond the minimum perimeter of the glacier could however also be determined and included in the computations. Or is there a reason that glacier extent has not been re-mapped every year?

**Response**

The annual glacier area is confirmed by the change of glacier terminus. However, we could not confirm the annual glacier area in the Japanese Alps, because snow cover around glaciers does not disappear on a positive balance year. In this case, we could not know the annual glacier outline in positive balance year. Therefore, we used the smallest glacier area for the calculation of glacier mass balance during the observation period. This area is composed of bare ice on the surface of the whole area. Therefore, this area is definitely a glacier area.

Line 159: Why is there only a 10m buffer around the glacier extent? The current glacier extent, or the minimum one? Chances are pretty high that there is remnant some snow close to the glacier in either the first or the second terrain model that would completely distort the correction process. Given that a terrain model can be generated for a bigger perimeter around the glaciers, why did the authors not use all of the stable terrain for this assessment? I would actually EXCLUDE the buffer zone. The text however clearly indicates that the comparison was done WITHIN the buffer zone. Furthermore, in Figure 4 it seems that the buffer is much bigger than the 10m stated. More explanation is needed here.

**Response**

In the case that we set a buffer zone around the glacier area, the DSM comparison includes the change of snow depth, because snow cover around glacier does not disappear on positive balance year. To avoid this, we took a 10 m buffer zone from the outline of the remaining snow area around the glacier in the positive balance year of 2017. The entire area within this buffer zone is the bedrocks. We compared the DSM differences at the same location through the observation period.

As the reason we do not calculate on all stable terrain, the accuracy of DSM depends on the slope (Fig. 7). The rock wall surrounding the glacier in the Japanese Alps is a steep slope. Therefore, the steep slope was excluded, to verify the accuracy in the slope zone of the same degree as the glacier area.

Line 161: probably "winter balance error" is meant here.

**Response**

We change it.

Line 169: Please use subscript for w and s after mass balance B (and not just Bw, Bs), always following the terminology proposed in Cogley et al., 2011.

**Response**

We change it.

Line 208: Do you mean "in elevation" instead of "in slope"?

**Response**

This is "in slope". The slope is steeper, the accuracy of the DSM becomes poor.

Line 222: This uncertainty refers to the digital elevation models but does not account for the (important) uncertainty due to unknown density of volume change. This aspect also needs to be considered.

**Response**

As responded second substantive comment above, in revise, we exclude the winter and summer balances in Table 4, Figures 9 and 13, and instead shown as accumulation depth and ablation depth. However, to compare with the mass balance amplitude of glaciers which is recorded in WGMS, we need to calculate the winter and summer balances of the northern Japanese Alps. In Japanese mountains, the snow density of avalanche deposits is larger than that of snowfall snow layers (Naruse et al., 1986; Abe et al., 2016). In the northern Japanese Alps, the density of avalanche deposits is 590 kg/m$^3$ (Shimizu et al., 1973). Therefore, we calculate the mass balance amplitude using the winter snow density in the range of 431 to 590 kg/m$^3$. Figures 11 and 12 and the appendix are also revised accordingly.

Line 247: The analysis of mass balance amplitudes is interesting. However, the unit of all numbers is wrong! Numbers are given in mm w.e., instead of m w.e. as stated in the text and in the figures. I would consistently convert all numbers to m w.e. (i.e. divide them by 1000). Furthermore, the effect of the partly short series (just four years for the studied glaciers) should be discussed. How strongly do the extreme years (2015-2016, high loss; 2016-2017: high gain) affect the result? Are the four years statistically sufficient to draw a final conclusion?

**Response**

We changed all units. The negative balance year of 2015-2016 and positive balance year of 2016-2017 are the largest and smallest snow depth respectively in Murodo-daira through the past 20 years. The total mass balance of both years was canceled out. The mass balance of VSGs was first observed in the Japanese Alps in this study. Our data showed that the glacier mass lost largely over a four-year period. The purpose of our paper is to discuss the mass balance changes over a four-year period. The characteristics of the glaciers in the Japanese Alps are well demonstrated even over a four-year period.

Line 261: Although I think the measurements likely do not capture the full range of actual emergence

velocities, the analysis is well-conducted given the difficulties of direct field observations. Nevertheless, it remains unclear what has actually been done to derive mass balance profiles from the elevation changes. Which values for the emergence velocity have been applied, and how have they been extrapolated over the entire glacier surface, and to other glaciers? More details are needed.

**Response**

As responded first substantive comment above, we change a profile of altitude changes instead of a mass balance profile in Figure 13, because accurate estimates of emergence velocities are difficult. In the Changri Nup glacier in Nepal, the emergence velocity of the glacier terminus is about 0.37 m a$^{-1}$ against an average horizontal velocity of 9.7 m a$^{-1}$ (Vincent et al., 2016). In the Argentière Glacier in the French Alps, the emergence velocity of the glacier terminus is 3-6 m a$^{-1}$ against a horizontal velocity of 35-60 m a$^{-1}$ (Vincent et al., 2021). As shown in these references, the emergence velocity is less than a tenth of the horizontal velocity on the glacier surface. In the case of VSGs in the Japanese Alps, the horizontal velocity is less than 4 m a$^{-1}$, and we consider that the emergence velocity is extremely small. In our revised paper, we add the above explanation with some references, to explain that the emergence velocity of VSGs in the Japanese Alps is extremely small. We discuss the characteristics of the mass balance profile using the profile of the altitude change as an estimation that the emergence velocity is low.

Line 308: Interesting observation. Any possible explanations?

**Response**

The northern Japanese Alps is a heavily snow-covered area in the world due to the existence of Siberian High and the warm Tsushima current (Kawase et al., 2020). Annual variations of snowfall are quite large, which has a significant impact on the accumulation of avalanches and snowdrifts. The annual change of snow depth also influenced the volume of avalanches and snowdrifts. Therefore, the annual mass balance of VSGs is greatly affected by annual changes in winter mass balance.

Line 313: Well, it is not the mass balance profile that has been measured but elevation change. Actually, typical alpine glaciers would show very similar profiles of elevation change over seasonal and annual periods (the latter only if their mass balance is close to zero)! However, accounting for emergence velocities leads to the reported mass balance profiles that are based on local measurements of mass balance. In that sense, I am still a bit reluctant to accept that emergence velocities are more or less zero throughout the entire elevation range and that mass balance profiles are flat on the investigated Japanese glaciers.

**Response**

As shown in these references (Vincent et al., 2016; 2021), the emergence velocity is less than a tenth of the horizontal velocity on the glacier surface. In the case of VSGs in the Japanese Alps, the

horizontal velocity is less than 4 m a$^{-1}$, and we consider that the emergence velocity is extremely small. We discuss the characteristics of the mass balance profile using the profile of the altitude change as an estimation that the appearance speed is low.

Line 339: It appears to me that, according to the Abstract of that paper, the stated number is incorrect.

**Response**

We change it.

Appendix: This very long table should rather go into a Supplementary Material and not an Appendix that is actually coming along in the same pdf as the paper. It is information that does not strictly need to be in paper, i.e. can also directly be obtained from the WGMS. Quickly state how the glaciers are ordered in the table. Units for winter and summer balance, as well as amplitude are wrong (mm w.e. instead of m w.e.).

**Response**

We change the unit and include to Supplementary Material.

**Reference**

Abe, O., Nakamura, K., Sato, K. and Kenji, K.: Observation of Frequent Avalanches in Sekiyama Pass along Route 48 and Assessment of Snowpack Stability on the Valley Slopes, Natural Disaster Research Report, (49), 39–46 [online] Available from: https://dil-opac.bosai.go.jp/publication/nied_natural_disaster/pdf/49/49-06.pdf, 2016.

Cogley, J. G., Hock, R., Rasmussen, L. A., Arendt, A. A., Bauder, A., Braithwaite, R. J., Jansson, P., Kaser, G., Möller, M., Nicholson, L. and Zemp, M.: Glossary of glacier mass balance and related terms, IHP-VII Technical Documents in Hydrology, 86, 114, doi:10.5167/uzh-53475, 2011.

Fukui, K., Iida, H. and Kosaka, T.: Newly Identifying Active Glaciers in the Northern Japanese Alps and Their Characteristics, Geographical Review of Japan Series A., 91(1), 43–61 [online] Available from: https://ci.nii.ac.jp/naid/40021445177/ (Accessed 22 March 2021), 2018.

Kawase, H., Yamazaki, T., Sugimoto, S., Sasai, T., Ito, R., Hamada, T., Kuribayashi, M., Fujita, M., Murata, A., Nosaka, M. and Sasaki, H.: Changes in extremely heavy and light snow-cover winters due to global warming over high mountainous areas in central Japan, Prog. Earth Planet. Sci., 7(1), 1–17, doi:10.1186/s40645-020-0322-x, 2020.

Kawashima, K.: Formation processes of ice body revealed by the internal structure of perennial snow patches in Japan, Bull. Glac. Res., (15), 1–10 [online] Available from: https://pascal-francis.inist.fr/vibad/index.php?action=getRecordDetail&idt=2764924 (Accessed 23 March 2021), 1997.

Naruse, R., Nishimura, K. and Maeno, N.: Studies on Mixed-phase Snow Flows IV : Stop and Accumulation Processes, Low temperature science. Series A, Physical sciences, 44, 165–176 [online] Available from: https://eprints.lib.hokudai.ac.jp/dspace/bitstream/2115/18527/1/44_p165-176.pdf, 1986.

Shimizu, H., Fujioka, T., Nakagawa, M., Kamada, K., Akimiya, E. and Narita, H.: Study of High-Speed Avalanche in Kurobe Canyon III, Low temperature science. Series A, Physical sciences, 32, 113–127 [online] Available from: https://eprints.lib.hokudai.ac.jp/dspace/bitstream/2115/18251/1/32_p113-127.pdf, 1974.

Vincent, C., Wagnon, P., Shea, J. M., Immerzeel, W. W., Kraaijenbrink, P., Shrestha, D., Soruco, A., Arnaud, Y., Brun, F., Berthier, E. and Others: Reduced melt on debris-covered glaciers: investigations from Changri Nup Glacier, Nepal, The Cryosphere, 10(4), 1845–1858 [online] Available from: https://tc.copernicus.org/preprints/tc-2016-75/, 2016.

Vincent, C., Cusicanqui, D., Jourdain, B., Laarman, O., Six, D., Gilbert, A., Walpersdorf, A., Rabatel,

A., Piard, L., Gimbert, F. and Others: Geodetic point surface mass balances: a new approach to determine point surface mass balances on glaciers from remote sensing measurements, The Cryosphere, 15(3), 1259–1276 [online] Available from: https://tc.copernicus.org/preprints/tc-2020-239/, 2021.

---

## Author Comment (AC2)

**Dear Dr Ian S. Evans**

**(corresponding to https://doi.org/10.5194/tc-2021-182-RC2 )**

We thank you for your valuable comments. Here we address how we will revise the manuscript corresponding to your specific comments. The comments are in black words, which are followed by our responses in red words.

**Comment**

The authors report very interesting observations on five Japanese glaciers, over several years. These are clearly exceptional compared with other glaciers, which is attributed to avalanching and perhaps wind drifting, combined with very heavy snowfall. They are narrow and linear. All the Tables and Figures are clear and informative. The writing is concise and the structure is acceptable. The expression is readable but occasionally imprecise, with a strange use of 'between', and there are a few careless errors.

More context would be welcome, both on other possible glaciers among the "More than 100 perennial snow patches "in this region, and on comparison with other avalanche-fed glaciers and other glacierets / very small glaciers elsewhere. Can the feeding avalanche tracks be mapped? Are there cornices in winter, suggestive of wind drifting? Many aspects of the literature are well covered, but a number of other papers on glacierets or very small glaciers could be considered:

Some of these state that mass balances tend to vary from year to year, positive throughout or negative throughout, rather than between accumulation and ablation areas.

On the 'Inventory of perennial snow patches…', the Higuchi et al. GeoJournal paper is only 8 pages.   Might it be worth citing the fuller (81-page) Atlas? –

Higuchi, K., Iozawa, T.:   Atlas of perennial snow patches in Central Japan. Water Res. Lab., Faculty of Science, Nagoya U., 81 pp., 1971.

Apart from the detailed corrections, and extending comparisons with avalanche-fed glacierets elsewhere, my main suggestion is to compare the (1.92 to 4.34) ratio of winter mass balance to local weq snowfall, with similar ratios for the other nine glaciers smaller than 0.11 km$^2$ in the Appendix (or consider similar ratios for balance amplitude).

**Response**

The following is a split response to the above comments.

i.      More context would be welcome, both on other possible glaciers among the "More

than 100 perennial snow patches" in this region, and on comparison with other avalanche-fed glaciers and other glacierets / very small glaciers elsewhere.

**Response**

Other possible glaciers:

There are no references about other possible glaciers. The references related to glaciers are only Fukui and Iida (2012), Fukui et al. (2018), and Arie et al. (2019). These are cited in this paper. As you referred to comparison with other snow patches, we progress to investigate the influence of topographic conditions are investigated. In addition, we have researched three perennial snow patches in fieldwork. In this paper, we add some sentences about the topographic condition, because we focused on glacier mass balance in this paper.

The comparison of other topographic-controlled VSGs:

Topographic-controlled VSGs have been reported mainly in the European Alps but also Rocky, Urals, Andes. Results of comparing Topographic-controlled VSGs in this paper, we show that (1) the Japanese VSGs have the annual mass balance greatly affected by the winter balance, and (2) the mass balance amplitude of the Japanese VSGs is larger than that of any other VSGs in the world including glaciers located in a warmer and wetter region. As you pointed out, we did not mention that the mass balance of some VSGs becomes positive or negative throughout large annual fluctuations. In our revised paper, we include Colucci et al. (2021), a recent study of VSGs in the European Alps, and Fukui et al. (2018), which describes the same characteristics as VSGs in the European Alps.

ii.     Can the feeding avalanche tracks be mapped? Are there cornices in winter, suggestive of wind drifting?

**Response**

In aerial images, fresh avalanche tracks can be observed on the glacier. However, it is impossible to determine where all avalanches occurred during winter. Conies also can be observed on the upper ridge of the glacier. The previous study shows that avalanches and snowdrifts have effects on the formation of perennial snow ravines (Higuchi, 1968). In our revised paper, we add Higuchi (1968) and the effect of snowdrift on the east slope of the main mountain ridge and observation by aerial images.

iii.    On the 'Inventory of perennial snow patches…', the Higuchi et al. GeoJournal paper is only 8 pages.    Might it be worth citing the fuller (81-page) Atlas?

**Response**

As your points, we add the reference of Higuchi and Iozawa (1971).

iv.     my main suggestion is to compare the (1.92 to 4.34) ratio of winter mass balance to local weq snowfall, with similar ratios for the other nine glaciers smaller than 0.11 km$^2$ in the Appendix (or consider similar ratios for balance amplitude).

**Response**

This is a very interesting proposal, but it is difficult to compare it with other VSGs, because we do not have the snowfall amount data at the observation of winter balance.

The following is an answer to the DETAILS.

**DETAILS:**
'likely' should usually be replaced by 'probably'.
**Response**
We replace all 'likely' with 'probably' in the sentence.

Line 19 Why not 'very small avalanche-fed glaciers'?    'Topographically controlled' is a broader class, including effects of aspect (shade) and shadow, and shelter from wind drifting snow off a plateau (not in this rugged part of the Japanese Alps!).
**Response**
The VSGs in the Japanese Alps are located at the valley bottom in the eastern part of the mountain range. This means an effect of avalanche, snowdrift, and shade. Some of the VSGs in the Japanese Alps are covered by debris. In our revised paper, we add more evidence and effects to classify them as topographic-controlled VSGs.

69 presumably end-winter: best to give a date for this, here.
**Response**
We give a date.

73-78 'glacial erosion valleys' are usually termed
**Response**
We replace 'glacial erosion valleys' with 'glacial troughs'.

82 'with little debris'
**Response**
We change it.

Table 1 Karamatsuzawa cannot be 1.03 km$^2$.    That contradicts the Introduction and all the maps.
Also, the average inclination is sometimes close to (altitude range)/ length, but not for Komado and Karamatsuzawa – i.e. it is not overall inclination.    Perhaps define how it was calculated.
**Response**
We mistook the unit of measurement for Karamatsuzawa Glacier. The correct value is 0.103 km$^2$. This inclination angle was calculated on ArcGIS using the DSM created by

aerial images and SfM software. This inclination angle is the average value. However, this method is affected by crevasses and ice holes. This may be the reason why the average slope does not match the (altitude range)/length. Therefore, as your suggestion, we calculate the average inclination of all glaciers using the altitude range and length($\theta = tan^{-1} b/a$). We also improve Table 1.

95-96 'from an altitude range'
**Response**

We change it.

155-155 How can you be so precise, unless photos were taken daily?
**Response**
We used the smallest area for each glacier at the end of the snowmelt season in observation periods.
We add "for five years" to the end of the previous sentence in Line 153.

158 'between … and …' ??
**Response**
We change it.

Fig.4 Why does the '10 m' buffer zone extend much further, especially downglacier in c and d, and upglacier in a ?
**Response**
In the case that we set a buffer zone around the glacier area, the DSM comparison includes the change of snow depth, because snow cover around glacier does not disappear on positive balance year. To avoid this, we took a 10 m buffer zone from the outline of the remaining snow area around the glacier in the positive balance year of 2017. The entire area within this buffer zone is the bedrocks. We compared the DSM differences at the same location through the observation period.

Fig.5 caption Not 'Each circle represents' – delete that. Rather 'The following numbers of glaciers in each region are included: …' [ thereafter, repetition of 'glaciers' is unnecessary]

**Response**

We change it.

179 'amounts'

**Response**

We change it.

180 'accumulation increases with'

**Response**

We change it.

Fig.7 on left: 'slope'

**Response**

We change it.

186-187 The sentence seems redundant, if you just mention 'profile and gradient' earlier.

**Response**

In the first part of this chapter, we mentioned only the profile, but did not explain the mass balance gradient. We explain that the slope of the profile is a mass balance gradient. For this reason, we remain this sentence.

219 should it be 'within' rather than 'between'?

**Response**

In this sentence, "between" is correct.

Because we calculated the altitude difference between DSM and GPS.

221 again unsure how 'between' is being used.

**Response**

We change it.

230 & 234: '5 to 13 m', not 5 to 11 m. (From Table 4: winter 5.63 to 12.72, summer 7.16 to 11.64.)   Per glacier, summer balances vary up to 1.6 m, so are not exactly constant, just with much smaller variation than for winter.

**Response**

We change it.

238 'nearly constant' is an exaggeration. I would say 'is much less variable'

**Response**

We change it.

243 Add 'It is reassuring that balances are closely correlated between the five glaciers, over this time period (Figs. 9 and 10).'

**Response**

We add it.

Table 3: seems like 'correction' should read 'corrected'. Caption should remind reader what the correction is for. (The text does not even mention correction, except for lines 111-114 on the DSM: is that relevant, or is the correction for emergence velocity?)

**Response**

We change it.

This correction is calculated from the mean and standard deviation of the DSM differences within the buffer zone on the bedrock. We subtract the mean value of the difference in the buffer zone from the value of the difference on the glacier. The standard deviation of the buffer zone is used as the error (±). We add this sentence to the caption.

251-252. Yes, but it is more logical to say that standard deviations increase linearly with amplitudes.

**Response**

We change it.

Fig.11 Not m ! Numbers on y axis are in mm, unless 000 is deleted throughout.

**Response**

We change it.

255-262 As emergence velocity has presumably been used above, should this section come earlier? More explanation of how it was used is needed.

**Response**

In the case of calculation of the profile using the geodesic method, the emergence velocity is needed to calculate. In the previous section about the mass balance of the entire area, the emergence velocities of the upstream part and downstream part are canceled out by each other. Therefore, we do not need to calculate it in the previous section. The position

of this section is correct, because this calculation is necessary for the profile.

Fig. 11 provides an interesting comparison between regions, but is not the most relevant way of comparing mass balance data with the results in Japan. Clearly glacier size is important: the larger the glacier, the less important the topographic effects including avalanching and wind drifting of snow. As the Japanese glaciers are 0.11 km2 or smaller, I suggest focussing on comparison with the small number of glaciers in the Appendix which are also of such tiny areas. I think there are nine: numbers 20, 21, 27, 51, 56, 74, 139, 151 and 180. One is in New Zealand, three in the Andes, four in the Alps and one in Apennines (Calderone is not in the Alps.) As the winter balances in Japan are 1.92 to 4.34 times the average direct snowfall (2.93 m), it would be useful to calculate similar ratios for the other very small glaciers. I think that would reveal that glaciers e.g. in the Urals have similar ratios, from what has been termed 'suralimentation' ('over-feeding'). That is, the Japanese glaciers are exceptional (practically 'outliers') in absolute terms, but probably not in amplitude relative to direct snowfall. (More accurate ratios could be calculated from annual snowfall values, rather than e.g. average snowfall at Tateyama Murododaira.)

**Response**

Fig. 12 shows that the VSGs in the Japanese Alps have the largest mass balance amplitude among all glaciers which is recorded in WGMS, including VSGs. Fig. 11 is the same result as Fig. 12. In our revised paper, we exclude Fig. 11, and show VSGs in the Japanese Alps have the largest mass balance amplitude using Fig. 12.

We are interested in your suggestion (Comparison of the ratio of snowfall to winter balance in VSGs), but we do not have local precipitation data of the same elevation. If you have some data or know data site, please let me know.

Table 5: 'a-1'

**Response**

We change it.

275 'from typical glaciers,' - although I would rather say 'from valley glaciers' because many cirque glaciers also have much snow-drift and avalanche input.

**Response**

We change it.

293-2895 Indeed: the dependence of Ural glaciers on wind-drifting (from summit

plateaus) and consequent avalanching was noted long ago by Dolgushin:

**Response**

We add the reference.

295 Not narrow valleys, but cirques.

**Response**

We change it.

298-299 delete one of the repeated 'In addition,'s.

**Response**

We change it.

302 delete 'likely'?

**Response**

We change it.

313 'vertical profiles'

**Response**

We change it.

315 'become negative throughout' OR 'Even if the glacier can become an ablation-area throughout'

**Response**

We change to 'Even if the glacier can become an ablation-area throughout'.

325 It might be good to mention (somewhere) that (from the maps in Fig.3 and photos in Fig.2), the glaciers seem to receive avalanche snow throughout – not just at their upstream ends.   Also, the evidence for some wind-drifting effect (despite the lack of plateaus or even rounded summits – the ridges are rugged) comes from the eastward component of aspect of all five glaciers.

**Response**

In line 78, we add 'These glaciers are located at the valley bottom surrounded by steep bedrock in the east part of the mountain range. This means snow avalanches and snowdrifts contribute largely to the accumulation of the entire glacier'. In fact, most glaciers and snow patches are located on the east part.

336 'tend to have annual balances strongly …'

**Response**

We change it.

349 Yes, very probably, but by how much?

**Response**

Kawase et al. (2020) showed that an increase of 2 °C lead to about 10% decrease in the maximum snow cover in light snow years and about 30% decrease in heavy snow years. We could not show the value because the article did not show a specific value.

349 '2°K'

**Response**

We change it.

354 delete first 'the'.

**Response**

We change it.

359 'very little' - not 'almost no'.

**Response**

We change it.

362 'of a typical'

**Response**

We change it.

364 'in Japan often had negative winter balance gradients'.    Fig. 13 does not really show negative annual balance gradients, and line 281 states "the gradients vary significantly".    OK they did not have ELAs: that is because of the lack of positive annual mass balance gradients.

**Response**

We change to 'they did not have ELAs: that is because of the lack of positive annual balance gradients'

375 Appendix A:    Another 3-orders-of-magnitude error: clearly the amplitude for #1 cannot be 10.7 km. The 4 columns ABw – SBn must be in mm, not m!

**Response**

We change it.

391 'fee' ??

**Response**

We change it.

406 'Ödenwinkelkees, central Austria,'

**Response**

We change it.

429 '4(4), 303-311'

**Response**

We change it.

439 'The Cryosphere'

**Response**

We change it.

**Reference**

Arie, K., Narama, C., Fukui, K., Iida, H. and Takahashi, K.: Ice thickness and flow of the Karamatsuzawa perennial snow patch in the northern Japanese Alps, Journal of the Japanese Society of Snow and Ice, 81(6), 283–295, doi:10.5331/seppyo.81.6_283, 2019.

Colucci, R. R., Žebre, M., Torma, C. Z., Glasser, N. F., Maset, E., Del Gobbo, C. and Pillon, S.: Recent Increases in Winter Snowfall Provide Resilience to Very Small Glaciers in the Julian Alps, Europe, Atmosphere , 12(2), 263, doi:10.3390/atmos12020263, 2021.

Fukui, K. and Iida, H.: Identifying active glaciers in Mt. Tateyama and Mt. Tsurugi in the northern Japanese Alps, central Japan, Journal of the Japanese Society of Snow and Ice, 74, 213–222, 2012.

Fukui, K., Iida, H. and Kosaka, T.: Newly Identifying Active Glaciers in the Northern Japanese Alps and Their Characteristics, Geographical Review of Japan Series A., 91(1), 43–61 [online] Available from: https://ci.nii.ac.jp/naid/40021445177/ (Accessed 22 March 2021), 2018.

Higuchi, K.: A Review of the Glaciological Studies on Perennial Ice and Snow Masses in Japan, Journal of the Japanese Society of Snow and Ice, 30(6), 195–207, doi:10.5331/seppyo.30.195, 1968.

Higuchi, K. and Iozawa, T.: Atlas of perennial snow patches in central Japan, Water Research Laboratory. Faculty of Science, Nagoya University., 1971.

Kawase, H., Yamazaki, T., Sugimoto, S., Sasai, T., Ito, R., Hamada, T., Kuribayashi, M., Fujita, M., Murata, A., Nosaka, M. and Sasaki, H.: Changes in extremely heavy and light snow-cover winters due to global warming over high mountainous areas in central Japan, Prog. Earth Planet. Sci., 7(1), 1–17, doi:10.1186/s40645-020-0322-x, 2020.

---

## Author Response (AR3)

Dear Chris,

Thank you for your proofread and valuable comments. Here we address how we revise the manuscript corresponding to your specific comments. The comments are in black words, followed by our responses in red words.

Moreover, to improve the manuscript, it was edited in English by Redmond Physical Sciences. The English edited manuscript was made English-ready for publication in this journal without altering the scientific content.

Kind regards,

Kenshiro Arie.

**Your comments**

Line 24: might be better as "These VSGs lack [or lacked] a positive annual mass balance gradient..."

**Response**

We change it.

Line 27: might be better as "Moreover, compared to other glaciers worldwide..."

**Response**

We change it.

Line 30: might be better as "...as determined by their density..."

**Response**

We change it.

Line 32: might be better as "Recently, however, smaller and more accurate surveying instruments..."

**Response**

We change it.

Line 50: delete comma after "glaciological"

**Response**

We change it.

Line 51: In the new sentence, make it clear "throughout" what? The mass balance year or...?

**Response**

We change as "Based on the mass balance, they argued that the entire glacier area was an accumulation area in 2012-2015, but an ablation area in 2015-2016."

Line 78: might be clearer as "observed during December 2010, and in January, February and March 2011, by the..."

**Response**

We change as "For meteorological data, we refer to that from a nearby station on Mt. Hakubadake (2,932 m), run by the Research Center for Mountain Environment of Shinshu University. Here, the maximum daily windspeeds for the months of December 2010 through March 2011 are 23.5, 34.9, 23.3, and 25.9 m s-1 respectively with wind directions are mainly west and northwest."

Line 82: might be clearer as "The average March snowdepth during 1996-2018 at..." [and then edit the rest of the sentence accordingly]

**Response**

We change as "Moreover, the average March snow depth at Murododaira (~2,450 m; Fig. 1), located at the west side of the mountain ridge, was about 6.8 m during 1996–2018 (Iida et al., 2018)."

Line 95: "These glaciers are located within the base of the troughs and surrounded..."

**Response**

We change this sentence as "Each glacier sits at the base of a trough just east of a mountain ridge, surrounded by steep bedrock, meaning that snow avalanches and snowdrifts can greatly contribute to the accumulation of the entire glacier."

Line 107: "Images were acquired on 9 Oct... [etc.]"

**Response**

We change it.

Line 166: "The accumulation and ablation are calculated by one integrates..." - this does not make sense grammatically.

**Response**

We change as "The accumulation depth is calculated by dividing the relative volume change during winter by the entire glacier area. For ablation, we instead use that during summer."

Line 201: delete "that"

**Response**

We change it.

Section 4.4: suggest you use either m/a or m a-1 (with the -1 superscript). Check journal preference/convention and the rest of the manuscript.

**Response**

We use only m a$^{-1}$ as in Mathematical notation and terminology of TC.

Line 319: change to "(see section 4.5)."

**Response**

We change it.

Line 369: I think this should be "too much snow" but what do you mean by "too much". Please clarify or re-write the sentence.

**Response**

We change as "Therefore, steep bedrock walls surrounding the VSGs in the northern Japanese Alps cause avalanches that supply more snow to the trough bottom than glaciers in the Urals."

Line 370: see previous point.

**Response**

We change as "In addition, strong north-westerly wind during winter in the northern Japanese Alps supply much snow to the east part of the mountain ridge due to snowdrift."

Line 442: "VSGs in Japan lack a positive annual mass..."

**Response**

We change it.

Line 447: "probably due to a warm mid-latitude climate with very heavy snowfall, supplemented by avalanches and snowdrift..."

**Response**

We change as "probably due to being in a warm mid-latitude climate with very heavy snowfall, supplemented by avalanches and snowdrift"

Line 451: "if the amount of snow in Japan were to decrease due to climate change..." - this is an intriguing and potentially important scenario but you give no indication as to how likely it is and what climate projections indicate. Can you cite any work suggesting this is likely or possible (or unlikely) for the interested reader? This may help increase the impact of your conclusions.

**Response**

We add "as predicted by climate models (Kawase et al., 2020)", however, this had already been shown in the discussion 5.2.